# Prevalence of dermatological toxicities in patients with melanoma undergoing immunotherapy: Systematic review and meta-analysis

**Náthali Felícia Mineiro dos Santos Garrett**[1☯], **Ana Cristina Carvalho da Costa**[2☯], **Elaine Barros Ferreira**[1‡], **Giovanni Damiani**[3,4,5,6‡], **Paula Elaine Diniz dos Reis**[1‡], **Christiane Inocêncio Vasques**[1☯]*

1 School of Health Sciences, University of Brasília, Brasília, Brazil, 2 Department of Health of the Federal District, Brasília, Brazil, 3 Clinical Dermatology, IRCCS Istituto Ortopedico Galeazzi, Milan, Italy, 4 Department of Biomedical, Surgical and Dental Sciences, University of Milan, Milan, Italy, 5 Department of Dermatology, Case Western Reserve University, Cleveland, OH, United States of America, 6 Department of Drug Sciences, University of Padua, Padua, Italy

☯ These authors contributed equally to this work.
‡ These authors also contributed equaly to this work.
* chvasques@unb.br

## Abstract

### Background

Checkpoint inhibitors have revolutionized advanced melanoma care; however, their cutaneous side effects have not been definitively elucidated.

### Objective

To identify the prevalence of cutaneous toxicity in patients with melanoma treated with immune checkpoint inhibitors as monotherapy and/or in combination with chemotherapy and/or radiotherapy.

### Materials and methods

We performed a systematic review and meta-analysis, which encompassed both clinical trials and observational studies describing the dermatological toxicities in patients treated with immune checkpoint inhibitors. The protocol was registered in the International Prospective Register of Systematic Review under the number CRD42018091915. The searches were performed using the CINAHL, Cochrane CENTRAL, LILACS, LIVIVO, PubMed, Scopus, and Web of Science databases. The methodological quality of the studies was evaluated with the JBI Critical Appraisal Checklist for Studies Reporting Prevalence Data

### Results

A total of 9,802 articles were identified in the databases. The final sample comprised 39 studies. The evaluated drugs were ipilimumab, tremelimumab, pembrolizumab, and

**Data Availability Statement:** All relevant data are within the paper and its Supporting Information files.

**Funding:** This work was funded by a grant from the National Council for Scientific and Technological Development (CNPq), Brazil to CIV (Grant no. 437802/2018-3). The funders had no role in study design, data collection and analysis, decision to publish, or preparation of the manuscript.

**Competing interests:** The authors have declared that no competing interests exist.

nivolumab. The results suggest that the most prevalent side effect was grade 1 and 2 pruritus (24%), followed by grade 1 and 2 rash (21%) and grade 1 and 2 vitiligo (10%).

## Conclusion

The most prevalent side effects in patients treated with checkpoint inhibitors are pruritus, rash, and vitiligo, and they are rated mostly as grades 1 and 2 adverse events. Remarkably, vitiligo is most commonly found in patients treated with PD-1 inhibitors.

## Introduction

Immune checkpoint inhibitors (ICIs), which were originally introduced for the treatment of melanoma during the last decade, have revolutionized cancer therapy [1,2]. Many patients have been living longer due to remarkable responses and delay of disease progression during ICI treatment [3].

Unlike monoclonal antibodies, ICIs act as co-stimulatory inhibitory receptor antagonists to counteract the deactivation of the immune system caused by the tumor and to promote immune activation [4,5]. The majority of ICIs act on the inhibition of cytotoxic T-lymphocyte-associated antigen-4 (CTLA-4), programmed cell death protein 1 (PD-1), and programmed death-ligand 1 (PD-L1) [6].

Although in many cancers one can see anti-tumor activity with ICIs and traditional chemotherapy, the types, mechanisms, and rates of side effects differ [7,8]. The side effects related to ICIs are labeled as immune-related adverse events (irAEs) and are thought to be related to the inflammatory response caused in several organs due to the stimulation of the immune system, especially of T cells [9–11]. Though one can see irAEs involving all body systems, cutaneous toxicity is of particular interest.

The dermatological toxicity of ICIs is similar, although its incidence is higher with ipilimumab than with anti-PD-1 or anti-PD-L1 agents [12,13]. Cutaneous adverse events (AEs) attributed to CTLA-4 inhibitors usually occur within 3–6 weeks after the initiation of therapy. However, these AEs occur within 2–10 months with PD-1 and PD-L1 inhibitor therapy [14].

The cost associated with the management of dermatological toxicity in patients with metastatic melanoma reaches US $ 21,726.00 per month, which represent the total adjudicated amount paid to all providers for inpatient and outpatient services and drugs [15]. Therefore, understanding the risks for dermatological toxicity can help to identify cutaneous AEs early on and thus to enable a more assertive clinical management, in addition to reducing costs.

Previous systematic reviews have assessed the patterns of irAEs and the safety of one or more ICIs [1,16–19]. However, several previous studies limited the search of references in the databases to a specific period [1,16,17,19], whereas another study restricted the review to only a few ICIs [18].

This review is a comprehensive report on the prevalence of dermatological toxicity in patients with melanoma using ICIs as monotherapy and/or in combination with chemotherapy and/or radiotherapy.

## Materials and methods

### Protocol and registration

This systematic review was elaborated according to the Preferred Reporting Items for Systematic Reviews and Meta-Analyses Checklist (PRISMA) [20]. The protocol was registered in the

International Prospective Register of Systematic Review (PROSPERO) [21] under the number CRD42018091915.

## Eligibility criteria

This review aimed to answer the following guiding question, based on the PECO strategy—Population, Exposure, Control and Outcomes: "What is the prevalence of dermatological toxicity (O) in patients with melanoma (P) undergoing treatment with ICIs (E)?"

We included clinical trials (randomized and non-randomized) and observational studies that evaluated melanoma cancer patients undergoing treatment with a single ICI, a combination of ICIs, or a combination of an ICI with chemotherapy and/or radiotherapy and that described the prevalence of dermatological toxicity. There were no restrictions regarding the language or publication period.

The studies were analyzed for inclusion and exclusion criteria in two phases. In phase 1 (screening of the titles and abstracts), we excluded studies that evaluated children and adolescents with cancer; adult oncology patients treated with ICIs with an associated autoimmune disease; and adult oncology patients treated with ICIs who also received vaccines, target therapy, or other therapies. Further, we excluded studies with cutaneous toxicity not associated with the use of ICIs, studies showing other (non-dermatological) toxicities associated with the use of ICIs, literature reviews, letters, case reports, personal opinions, conferences, abstracts, and book chapters.

In phase 2 (full text reading), we excluded studies that presented complementary data from previously published investigations, qualitative studies, studies that did not present complete data and/or did not allow the extraction of the data of interest, and studies that evaluated other types of cancer (not melanoma).

## Information sources

The searches were carried out in the following electronic databases: CINAHL, Cochrane CENTRAL, LILACS, LIVIVO, PubMed, Scopus, and Web of Science. A search strategy was developed for each of the databases (S1 File). The gray literature was also checked using Google Scholar and Open Gray. Further, a manual search was performed on the list of references of the included studies. All searches were performed on January 23, 2019.

## Study selection

The study selection was carried out in two phases. In phase 1, two reviewers (N.F.M.S.G. and A.C.C.C.) independently assessed the titles and abstracts of all citations identified in the electronic databases. Articles that did not meet the inclusion criteria were excluded. This step was performed on a Rayyan web application [22]. In phase 2, the same reviewers applied the inclusion criteria to the full text of the articles. Any disagreements in the first or second phase were resolved by discussion until an agreement was reached between the two authors. When consensus could not be reached, the third reviewer (C.I.V.) was asked to evaluate the article and make the final decision.

References, including the removal of duplicates, were managed in EndNoteBasic [23].

## Data collection process

Two reviewers (N.F.M.S.G. and A.C.C.C.) independently collected data from the selected studies. A third reviewer (C.I.V.) evaluated the accuracy of the collected information. For all included studies, the following information was recorded: study characteristics (author, year,

country of publication, study design, and purpose); sample characteristics (sample size and mean age of the participants); duration of the drug treatment and follow-up; and characteristics of the results (type of dermatological toxicity and main conclusions).

### Risk of bias in individual studies

To evaluate the risk of bias, the JBI Critical Appraisal Checklist for Studies Reporting Prevalence Data [24], composed of nine items, was used. For each of the items, it was possible to check "yes", "no", "unclear", or "not/applicable".

The methodological quality of the studies was categorized as low risk of bias (70% or more "yes" responses), moderate risk of bias (50%–69% "yes" responses), and high risk of bias (up to 49% "yes" responses).

### Summary measures

The frequency of dermatological toxicity described by the studies was considered the primary outcome. The frequency was expressed as a percentage, which corresponds to the number of cases present among all patients treated with ICIs. We analyzed the toxicities with highest prevalence in the studies, allowing associative measures such as rash, pruritus, and vitiligo.

### Synthesis of the results

The meta-analysis was performed using the Jamovi software, version 1.6, which offers a variety of statistical techniques [25]. The prevalence was estimated by the number of events out of the total sample. We considered the homogeneity of the studies in relation to the type of exposure and outcome. Heterogeneity was calculated using $I^2$. An $I^2$ value greater than 50% was considered a substantial indicator of heterogeneity among the studies, and a random effect model could be used. When $I^2$ is less than 50%, a fixed effect model is recommended. Jamovi provides fixed and random effect models for each analysis, and we therefore chose between the two based on the $I^2$ values. Beside the $I^2$ values, we also considered the Confidence Interval and the p value. The level of significance was set at 0.05. Funnel plot was not considered, since it is a prevalence systematic review, and the graphic results could be non interpretability [26].

## Results

### Study selection

In phase 1, 9,802 studies were identified in the seven electronic databases. Among the 7,484 articles that remained after the removal of duplicates, 218 were selected. Another five studies that were identified in the gray literature were added and continued to phase 2. No additional study was identified in the list of references of the included articles.

Among the 223 studies that followed into Phase 2, 184 articles were excluded (S2 File). Thus, 39 studies [27–65] that met the eligibility criteria were included in the qualitative synthesis. Among these, 35 studies were considered adequate for the meta-analysis. The process of identification, inclusion, and exclusion of studies is described in Fig 1.

### Characteristics of the studies

The articles were published in English, except for one study, which was published in Spanish [39]. The publications occurred between 2008 and 2018, with the highest number of publications in 2017. There were predominantly multicenter studies (n = 13) [28,30,32,36,41,45–48,51,55,60,61] and studies conducted in the United States (n = 11) [27,29,31,33,35,37,38,41,42,56,64]. With respect to the study design, there were 28 clinical trials (randomized or

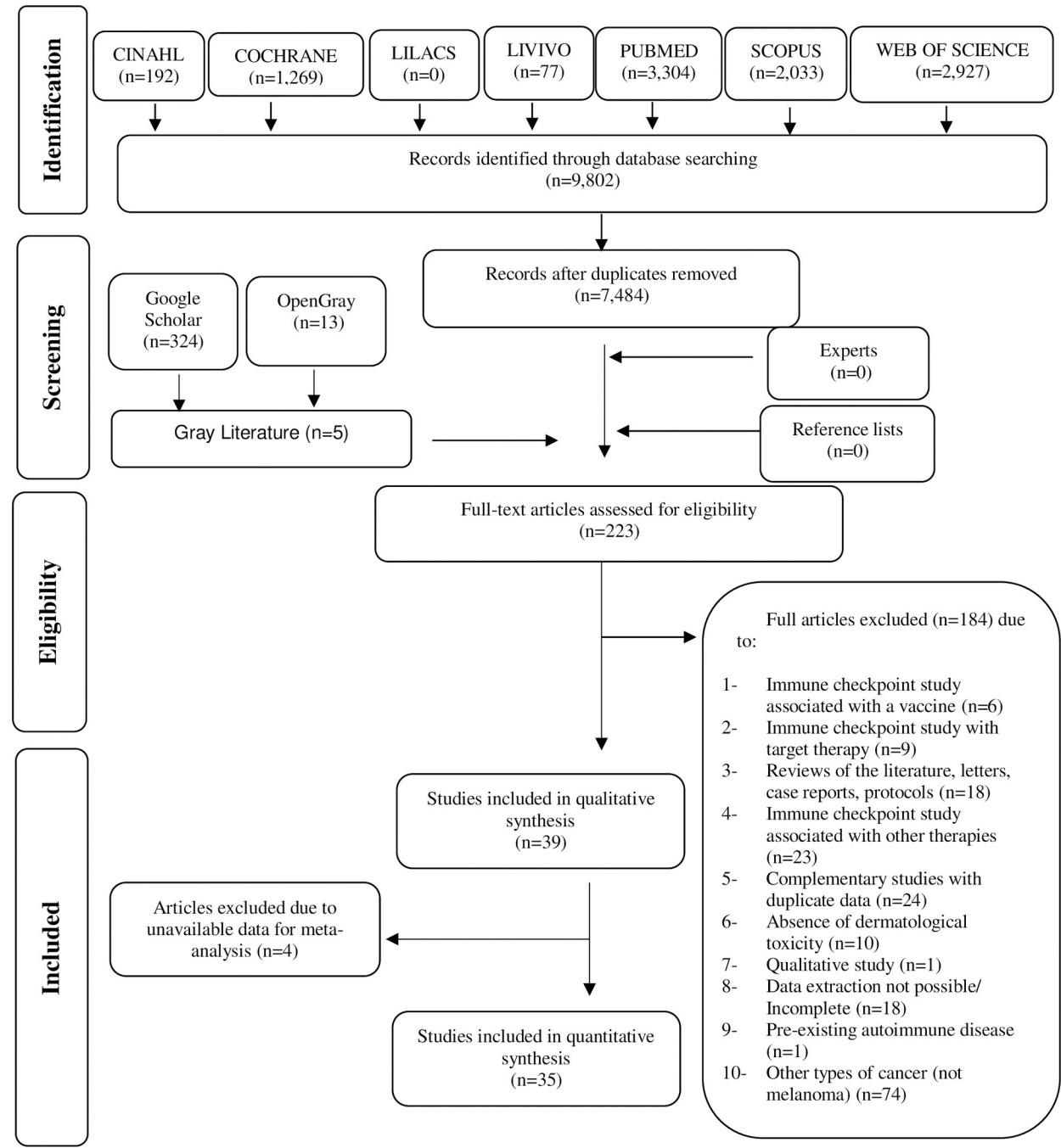

**Fig 1. Flow diagram of the literature search and selection criteria.** Adapted from PRISMA.

non-randomized) [27–31,33,35,37,38,40–48,51,55,56,58–64] and 11 observational studies [32,34,36,39,49,50,52–54,57,65].

Regarding the treatment with ICIs, 23 studies evaluated the use of ICIs alone, including the use of tremelimumab (n = 1) [28], ipilimumab (n = 14) [27,29,31,32,34,36,39,44,45–47,51–53], nivolumab (n = 4) [50,54,59,60], and pembrolizumab (n = 4) [33,49,58,61]. Combinations of two ICIs were evaluated in eight studies, including the combinations ipilimumab + nivolumab

(n = 6) [38,41,48,63–65] and ipilimumab + pembrolizumab (n = 1) [57]. Finally, the combination or comparison of ICIs with chemotherapy was assessed by six studies (n = 6) [30,35,37,40,42,62]. The characteristics of the studies included in the review are described in Table 1.

## Results of the individual studies

Among the studies included, 21 of them evaluated patients with advanced melanoma [29,32–34,37,38,40,43,44,48,50,53,55–60,62–64], 11 evaluated patients with metastatic melanoma [27,28,30,31,35,36,39,45,46,52,65] and seven studies evaluated patients with either stage III or IV irresectable melanoma or stage III or IV melanoma resected [41,42,47,49,51,54,61]. Overall, 9,847 patients were evaluated, and the samples in the individual studies ranged from ten [39] to 1,019 patients [61]. One of the studies [65] considered the total number of AEs observed during the study period as representative of the sample, thus not having a sample per number of patients. The mean age of the study participants was 60 years.

Regarding the ICI classes, only the anti-PD-L1 class was not analyzed in the studies included in this review. However, 15 studies studies evaluated the anti-CTLA4 class [27–29,31,32,34,36,39,44–47,51–53] and eight evaluated the anti-PD-1 class [33,49,50,54,58–61]. Several studies assessed the combined treatment with ICIs [38,41,43,48,55–57,63–65], whereas other studies evaluated the treatment with ICIs in combination with or in comparison to chemotherapy [30,35,37,40,42,62].

As to anti-CTLA4 class, the drug tremelimumab was evaluated alone at doses of 3, 6, 10, or 15 mg/kg [28] and, in comparison with chemotherapy (dacarbazine) [35]. Ipilimumab monotherapy was evaluated at doses of 3, 10, and 20 mg/kg, and the 3 mg/kg and 10 mg/kg doses were compared with each other [27,29,31,32,34,39,44–47,51–53]. The effect of ipilimumab treatment was also assessed in combination with nivolumab [38,41,48,63–65] and pembrolizumab [55,57], in comparison to nivolumab [56] or pembrolizumab [43] treatment, and in combination with chemotherapy (dacarbazine or paclitaxel + carboplatin) [30,37]. One study [36] did not describe the administered ipilimumab dose. Dermatological toxicity was more frequently observed in patients who used ipilimumab than in those who used nivolumab or pembrolizumab and included mainly rash and pruritus. Moreover, when the administration of ipilimumab was investigated in combination with chemotherapy, the dermatological toxicity was high in the ipilimumab group [30,37].

Regarding the anti-PD-1 class, the drug nivolumab was evaluated as monotherapy at doses of 2 mg/kg and 3 mg/kg [50,54,59,60] and compared with chemotherapy [40,62]. Higher frequencies of the AEs were observed at higher doses. When compared with chemotherapy, it was observed that the group treated with nivolumab also had dermatological AEs more frequently than the chemotherapy group in both grades 1 and 2, and in grades 3 and 4. Pembrolizumab was evaluated as monotherapy at doses of 2, 10, and 200 mg [33,49,58,61] and compared with chemotherapy [42].

Among the studies that were evaluated, the most frequent dermatological toxicities were rash [27–33,35,37–48,51,53–57,59–65], pruritus [28–33,35–46,48–49,51,53–65], vitiligo [27,33,38,40–43,49,50,52,55,57–62], dry skin [42,58,62], erythema multiforme [45,46], and skin hypopigmentation [58,59].

Other dermatological toxicities were reported by the studies such as alopecia [44], blister [38], dermatitis [32], drug eruption [55], eczema [52], erythema nodosum [51], folliculitis [52], hand-foot syndrome [45], leukoderma [59], lichenoid exanthema [36], maculopapular exanthema [36], mucositis [52], pemphigoid lesion [55], photosensitivity reaction [36], pruritic eczema [36], pyoderma gangraenosum [36], rosacea [52], seborrheic dermatitis [59],

**Table 1. Studies included in the review according to methodological characteristics (n = 39).**

| Author, Year Country | Study design | Sample size (n) | Age in years (mean and range) | Drug (dose and schedule) | Dermatological toxicities (n) % | |
|---|---|---|---|---|---|---|
| ALTOMONTE et al, 2013 [32] Multicentric | Retrospective OS | 74 | 56 (23–79) | IPI 10 mg/kg administered intravenously over 90 min, every 3 weeks, for a total of four doses Maintenance therapy: IPI 10 mg/kg every 12 weeks | Grade 1 or 2 Pruritus: (13) 17.6% Rash: (7) 9.4% Dermatitis: (1) 1.3% | |
| ASCIERTO et al, 2017 [51] Multicentric | RCT, phase 3 | 727 10 mg/kg group (365) 3 mg/kg group (362) | IPI (10 mg/kg): 62 (49–70) IPI (3 mg/kg): 62 (51–71) | IPI: 3 mg/kg IPI: 10 mg/kg Drug was administered by intravenous infusion for 90 min every 3 weeks for four doses | IPI (3 mg/kg) Grade 1or 2 Rash: (48) 13% Pruritic rash: (3) 1% Maculopapular rash: (4) 1% Pruritus: (79) 22% Grade 3 or 4 Rash: (2) 1% Pruritic rash: (1) <1% Pruritus: (2) 1% Erythema nodosum: (1) <1% Toxic skin eruption: (1) <1% | IPI (10mg/kg) Grade 1 or 2 Rash: (90) 25% Pruritic rash: (5) 1% Maculopapular rash: (3) 1% Pruritus: (80) 22% Toxic skin eruption: (1) <1% Grade 3 or 4 Rash: (5) 1% Maculopapular rash: (1) <1% Pruritus: (2) 1% |
| CAMACHO et al, 2009 [28] Multicentric | NRCT, phase 1 RCT, phase 2 | 117 28 (Phase 1) 89 (Phase 2) | Phase 1: not described Phase 2: 57.5 (20–83 | Tremelimumab Phase 1: at a dose of 3, 6 or 10 mg/kg (intravenous infusion once every 28 days) Phase 2: at a dose of 10 mg/kg (intravenous infusion monthly) or 15 mg/kg (intravenous infusion every 3 months) | Phase 1–3 mg/kg: Grade 1 or 2 Rash: (1) 33% Pruritus: (1) 33% Phase 1–6 mg/kg: Grade 1 or 2 Rash: (1) 33% Pruritus: (1) 33% Phase 1–10 mg/kg: Grade 1 or 2 | Rash: (10) 46% Pruritus: (10) 46% Phase 2–10 mg/kg: Grade 1 or 2 Rash: (15) 34% Pruritus: (13) 30% Phase 2–15 mg/kg: Grade 1 or 2 Rash: (16) 36% Pruritus: (15) 33% |
| DIKA et al, 2017 [52] Italy | Prospective OS | 41 | Not described | IPI at a dose of 3 mg/kg with 3 weeks interval. | Grade 1 or 2 Rash: (3) 7.2% Folliculitis: (3) 7.2% Folliculitis: (2) 4.8% Vitiligo: (2) 4.9% Mucositis: (1) 2.4% Rosacea: (1) 2.4% Eczema: (1) 2.4% Acneiform eruption: (1) 2.4% | Syringometaplasia mucinosa (1) 2.4% Grade 4 Stevens-Johnson syndrome: (1) 2.4% Not classification Pruritus: (4) 9.8% Xerosis: (2) 4.9% |
| EGGERMONT et al, 2016 [47] Multicentric | RCT, phase 3 | 951 IPI (475) Placebo (476) | 51.5 (18–84) | IPI (intravenously at a dose of 10 mg/kg every 3 weeks, for 4 doses) Placebo (every 3 weeks, for 4 doses) | IPI (n = 471) Grade 1 or 2 Rash: (156) 33.1% Any dermatologic event: (278) 59.02% Grade 3 Rash: (5) 1.1% Any dermatologic event: (20) 3.2% | Placebo (474) Grade 1 or 2 Rash: (52) 11,6% Any dermatologic event: (99) 20.8% Grade 3 Rash: 0 Any dermatologic event:0 |

(*Continued*)

**Table 1.** (Continued)

| Author, Year Country | Study design | Sample size (n) | Age in years (mean and range) | Drug (dose and schedule) | Dermatological toxicities (n) % | |
|---|---|---|---|---|---|---|
| EGGERMONT et al, 2018 [61] France | RCT | 1.019 Pembrolizumab (514) Placebo (505) | Pembrolizumab 54 (19–88) Placebo 54 (19–83) | Pembrolizumab (200mg) or Placebo intravenously every 3 weeks for a total of 18 doses (approximately 1 year) | Pembrolizumab Any Grade Rash: (82) 16.1% Pruritus: (90) 17.7% Severe skin reactions: (3) 0.6% Vitiligo: (24) 4.7% Grade >3 Rash: 1 (0.2) Severe skin reactions: (3) 0.6% | Placebo Any Grade Rash: 54 (10.8) Pruritus: 51 (10.2) Vitiligo: (8) 1.6% Grade <3 Rash (156) 33.1% |
| HAMID et al, 2013 [33] USA | NRCT | 135 | 60,4 (25–94) | Lambrolizumab (10 mg/kg) 30 min intravenous infusion, every 2 weeks, 2 mg/kg, 30 min intravenous infusion, every 3 weeks or 10 mg/kg, 30 min intravenous infusion, every 3 weeks. | Grade 1 or 2 Rash: (25) 18.5% Pruritus (27) 20% Vitiligo (12) 9% | Grade 3 or 4 Rash: (3) 2% Pruritus: (1) 1% |
| HODI et al, 2016 [48] Multicentric | RCT, phase 2 | 142 Nivolumab + IPI (95) IPI (47) | Not described | Nivolumab 1 mg/kg plus IPI 3 mg/kg or IPI 3 mg/kg plus placebo, every 3 weeks for four doses. Subsequently, patients assigned to nivolumab plus IPI received nivolumab 3 mg/kg every 2 weeks | Nivolumab + IPI Grade 1 or 2 Rash: (36) 38% Maculopapular rash: (12) 13% Pruritus: (37) 39% Grade 3 or 4 Rash: (4) 4% Maculopapular rash: (3) 3% Pruritus: (1) 1% | IPI Grade 1or 2 Rash: (14) 30% Maculopapular rash: (6) 13% Pruritus: (15) 33% |
| HUA et al, 2016 [49] France | Prospective OS | 67 | 54 (20–74) | Pembrolizumab (administered intravenously every 2 or 3 weeks at a dose ranging from 2 to 10 mg/kg). | Grade 1 or 2 Vitiligo: (17) 25% Pruritus: (16) 24% Eczematiform, lichenoid, or psoriasiform skin irritation: (18) 27% | |
| JUNG et al, 2017 [53] Korea | Retrospective OS | 104 | 58 (50–66) | IPI (intravenously at a dose of 3 mg/kg, every 3 weeks) 4 cycles. | Grade 1 or 2 Rash: (22) 21.1% Pruritus (32) 30.8% | Grade 3 Rash: (1) 1% Pruritus: (1) 1% |
| KU et al, 2010 [29] USA | NRCT | 51 | 62 (38–86) | IPI (intravenously at a dose of 10 mg/kg every 3 weeks, over 90 min, for 4 doses) maintenance ipilimumab 10 mg/kg every 12 weeks. | Grade 1 or 2 Pruritus: (22) 43% Rash: (18) 35% | Grade 3 Rash: (1) 2% |
| LARKIN et al, 2018 [62] England | RCT, phase 3 | 405 Nivolumab (272) ICC (133) | Nivolumab: 59 (23–88) ICC: 62 (29–85) | Nivolumab 3mg/kg intravenously every 2 weeks or ICC (DTIC 1,000 mg/m2 every 3 weeks or carboplatin area under the curve 6 plus paclitaxel 175 mg/m2 every 3 weeks) | Nivolumab Any grade Pruritus: (59) 22% Rash: (36) 13% Vitiligo: (29) 11% Maculopapular rash: (19) 7% Dry skin: (15) 6% Grade 3 or 4 Rash: (1) <1% Maculopapular rash: (1) <1% | ICC Any grade Pruritus: (1) 1% Rash: (5) 5% Maculopapular rash: (2) 2% |

*(Continued)*

**Table 1.** (Continued)

| Author, Year Country | Study design | Sample size (n) | Age in years (mean and range) | Drug (dose and schedule) | Dermatological toxicities (n) % | |
|---|---|---|---|---|---|---|
| LONG[a] et al, 2017 [54] Australia | Retrospective OS | 306 Non-beyond Progression (221) Beyond progression (85) | 62 (18–90) | Nivolumab (3mg/kg every 2 weeks) | Non-treatment beyond Progression (Non-TBP) Any Grade Pruritus: (25) 11% Rash: (23) 10% Grade 3 or 4 Pruritus: (1) <1% Rash: 0 | Treatment Beyond Progression (TBP) Any Grade Pruritus: (23) 27% Rash: (23) 27% Grade 3 or 4 Pruritus: 0 Rash: 0 |
| LONG[b] et al, 2017 [55] Multicentric | NRCT, phase 1b | 153 | 60 (53–70) | Pembrolizumab (2 mg/kg) intravenously for 30 min once every 3 weeks followed by IPI (1 mg/kg) intravenously for 90 min once every 3 weeks for four doses, followed by pembrolizumab 2 mg/kg intravenously for 30 min every 3 weeks for up to 2 years. | Treatment-related AEs Grade 1 or 2 Rash: (60) 39% Pruritus: (63) 41% Vitiligo: (30) 20% Rash Maculopapular: (18) 12% Pruritic rash: (7) 5% Macular rash: (6) 4% Drug eruption: (4) 3% Grade 3 or 4 Rash: (4) 3% Pruritus: 0 Rash Maculopapular: (1) 1% Pruritic rash: (2) 1% Macular rash: (1) 1% Drug eruption: (2) 1% DRESS syndrome: (1) 1% Pemphigoid: (1) 1% | irAEs Grade 1 or 2 Skin reactions: (1) 1% Grade 3 or 4 Skin reactions: (12) 8% |
| MARGOLIN et al, 2012 [31] USA | NRCT, phase 2 | 72 Cohort A (n = 51) Cohort B (n = 21) | Cohort A: 59 (33–79) Cohort B: 57 (30–74) | IPI (10 mg/kg) four doses, intravenous, one every 3 weeks. (designated weeks 1, 4, 7, and 10; induction). Patients who were clinically stable at 24 weeks were eligible to continue with treatment with IPI 10 g/kg every 12 weeks (maintenance). | Cohort A Grade 1 and 2 Rash: (17) 33% Pruritus: (16) 31% Grade 3 or 4 Rash: (1) 2% Pruritus: 0 | Cohort B Grade 1 and 2 Rash: (6) 29% Pruritus: (5) 24% Grade 3 or 4 Rash: (1) 5% Pruritus: 0 |
| RUIZ-MORALES et al, 2014 [39] Mexico | Retrospective OS | 10 | 49 (± 25) | IPI (3mg/kg) intravenous, during 90 min infusion every 3 weeks, with a total of 4 scheduled doses. | Grade 1 and 2 Pruritus: (3) 30% Rash: (2) 20% | Grade 3 and 4 Pruritus: 0 Rash: 0 |
| NAKAMURA et al, 2016 [50] Japan | Retrospective OS | 35 | 67 (40–85) | Nivolumab (intravenously at a dose of 2 mg/kg, every 3 weeks) | Grade 1 or 2 Vitiligo: (9) 25.7% | |

*(Continued)*

**Table 1.** (Continued)

| Author, Year Country | Study design | Sample size (n) | Age in years (mean and range) | Drug (dose and schedule) | Dermatological toxicities (n) % | |
|---|---|---|---|---|---|---|
| NAMIKAWA et al, 2018 [63] Multicentric | NRCT | 30 | 58.5 (31–81) | Nivolumab (1 mg/kg) plus IPI (3 mg/kg) every 3 weeks for four doses, followed by biweekly doses of nivolumab (3 mg/kg) | Grade 1 and 2 Rash: (18) 60% Pruritus: (10) 33% Rash maculopapular: (4) 13% | Grade 3 or 4: Rash: (2) 7% Pruritus: 0 Rash maculopapular: (1) 3% |
| POSTOW et al, 2013 [34] Multicentric | Retrospective OS | 33 25 (3 mg/kg) 8 patients (10 mg/kg) | 65 (35–90) | IPI (intravenously at a dose of 3 mg/kg, every 3 weeks or at a dose of 10 mg/kg). | IPI 3 mg/kg: Grade 1 and 2 Rash: (4) 15% | IPI 10 mg/kg: Grade 1 and 2 Rash: (2) 25% |
| POSTOW et al, 2015 [41] USA | RCT | 142 Nivolumab + IPI (95) IPI (47) | 65 (27–87) | IPI 3 mg/kg combined with either nivolumab 1 mg/kg or placebo every 3 weeks for 4 doses, followed by nivolumab 3 mg/kg or placebo every 2 weeks | Nivolumab + IPI Grade 1–2 Rash: (39) 41.5% Maculopapular rash (15) 16% Pruritic rash (3) 3.2% Pruritus: (33) 35.1% Vitiligo: (10) 10.6% Grade 3 or 4 Rash: (5) 5% Maculopapular rash (3) 3% Pruritus: (1) 1.1% | IPI Grade 1–2 Rash: (12) 26.1% Maculopapular rash (8) 17.4% Pruritic rash (5) 10.9% Pruritus: (13) 28.3% Vitiligo: (4) 8.7% |
| RIBAS et al, 2013 [35] USA | RCT, phase 3 | 655 Tremelimumab (328) Chemotherapy (327) | Tremelimumab: 57 (22–90) Chemotherapy: 56 (22–90) | Tremelimumab (15 mg/kg once every 90 days to four cycles) or DTIC (1,000 mg/m2) IV on day 1 of a 21-day cycle or single-agent Temozolomide (200 mg/m2) orally on days 1 to 5 of a 28-day cycle | Tremelimumab Any Grade Rash: (106) 33% Pruritus: (100) 31% Grade >3 Rash: (7) 2% Pruritus: (3) 1% | Chemotherapy Any Grade Rash: (17) 5% Pruritus: (16) 5% Grade >3 Rash: (1) <1% Pruritus: 0 |
| RIBAS et al, 2015 [42] USA | RCT, phase 2 | 540 Pembrolizumab 2 mg/kg (180) Pembrolizumab 10 mg/kg (181) Chemotherapy control (179) | Pembrolizumab 2 mg/kg: 62 (15–87) Pembrolizumab 10 mg/kg: 60 (27–89) Chemotherapy: 63 (27–87) | Pembrolizumab 2 mg/kg or 10 mg/kg every 3 weeks or investigator-choice chemotherapy (paclitaxel plus carboplatin, paclitaxel, carboplatin, DTIC, or oral temozolomide). | Pembrolizumab 2 mg/kg Grade 1 or 2 Pruritus: (37) 21% Rash: (21) 12% Vitiligo: (10) 6% Dry skin: (9) 5% Grade 3 or 4 Pruritus: 0 Rash: 0 Vitiligo: 0 Dry skin: 0 Pembrolizumab 10 mg/kg Grade 1 or 2 Pruritus: (42) 23% Rash: (18) 10% Vitiligo: (9) 5% | Dry skin: (9) 5% Grade 3 or 4 Pruritus: 0 Rash: 0 Vitiligo: 0 Dry skin: 0 Chemotherapy Grade 1or 2 Pruritus: (6) 4% Rash: (8) 5% Vitiligo: (2) 1% Dry skin: (2) 1% Grade 3 or 4 Pruritus: 0 Rash: 0 Vitiligo: 0 Dry skin: 0 |

(*Continued*)

**Table 1.** (*Continued*)

| Author, Year Country | Study design | Sample size (n) | Age in years (mean and range) | Drug (dose and schedule) | Dermatological toxicities (n) % | |
|---|---|---|---|---|---|---|
| ROBERT et al, 2011 [30] Multicentric | RCT, phase 3 | 502 IPI plus DTIC (250) Placebo plus DTIC (252) | IPI plus DT: 57.5 Placebo plus DTIC: 56.4 | IPI (10 mg/Kg) + DTIC (850 mg per square meter) or Placebo (given at weeks 1, 4, 7, and 10) + DTIC (850 mg per square meter) | | irAEs: IPI plus DTIC Total Pruritus: (66) 26.7% Rash: (55) 22.3% Grade 3 or 4 Pruritus: (5) 2% Rash: (3) 1.2% Placebo plus DTIC Total Pruritus: (15) 6% Rash: (12) 4.8% Grade 3 or 4 Pruritus: 0 Rash: 0 |
| ROBERT et al, 2014 [40] France | RCT | 418 Nivolumab (210) DTIC (208) | Nivolumab: 64 (18–86) DTIC: 66 (26–87) | Nivolumab (3 mg/kg of body weight every 2 weeks and DTIC-matched placebo every 3 weeks) or DTIC (1,000 mg per square meter of body-surface area every 3 weeks and nivolumab-matched placebo every 2 weeks) | Nivolumab Any grade Pruritus: (35) 17% Rash: (31) 15% Vitiligo: (22) 10.7% Grade 3 or 4 Pruritus: (1) 0.5% Rash: (1) 0.5% Vitiligo: 0 | DTIC Any grade Pruritus: (11) 5.4% Rash: (6) 2.9% Vitiligo: (1) 0.5% Grade 3 or 4 Pruritus: 0 Rash: 0 Vitiligo: 0 |
| ROBERT et al, 2015 [43] France | RCT | 834 Pembrolizumab every 2 Weeks (279) Pembrolizumab every 3 Weeks (277) IPI (278) | Pembrolizumab every 2 Weeks: 61 (18–89) | Pembrolizumab (at a dose of 10 mg/kg of body weight) every 2 weeks or every 3 weeks or four doses of IPI (at 3 mg/kg) every 3 weeks. | Pembrolizumab every 2 Weeks Any grade Rash: (41) 14.7% Pruritus: (40) 14.4% Vitiligo: (25) 9.0% Grade 3–5 Rash: 0 Pruritus: 0 Vitiligo: 0 Pembrolizumab every 3 Weeks Any grade Rash: (37) 13.4% Pruritus: (39) 14.1% Vitiligo: (31) 11.2% | Grade 3–5 Rash: 0 Pruritus: 0 Vitiligo: 0 IPI Any grade Rash: (37) 14.5% Pruritus: (65) 25.4% Vitiligo: (4) 1.6% Grade 3–5 Rash: (2) 0.8% Pruritus: (1) 0.4% Vitiligo: 0 |
| SHOUSHTARI et al, 2018 [64] USA | NRCT | 64 | 56 (22–82) | Intravenous nivolumab (1mg/kg) and IPI (3mg/kg) administered every 3 weeks for up to 4 doses, followed by nivolumab (3mg/kg) every 2 weeks or pembrolizumab (2mg/kg) every 3 weeks | Nivolumab + IPI Grade 1 or 2 Rash/pruritus: (13) 21% Grade 3 or 4 Rash/pruritus: (5) 8% Immune Related-AEs Rash/pruritus: (11 of 18) 61% | Anti- PD-1 Monotherapy Grade 2 Rash/pruritus: (2) 3% Grade 3 or 4 Rash/pruritus: 0 Immune Related- AEs Rash/pruritus: (10) 16% |

(*Continued*)

**Table 1.** (*Continued*)

| Author, Year Country | Study design | Sample size (n) | Age in years (mean and range) | Drug (dose and schedule) | Dermatological toxicities (n) % | |
|---|---|---|---|---|---|---|
| SOLDATOS et al, 2018 [65] Germany | Retrospective OS | 7,770 (number of AEs a certain occurrence was observed) | Not described | IPI only, Nivolumab only, IPI and nivolumab (dose and schedule, not described) | Nivolumab (n = 890 AEs) Rash: (38) 5.6% IPI (n = 2,704 EAs) Rash: (176) 6.5% Pruritus: (79) 2.9% | IPI and nivolumab (n = 682 AEs) Rash: (38) 5.6% Pruritus: (19) 2.8% |
| VOSKENS et al, 2013 [36] Multicentric | Retrospective OS | 752 | 60.1 (38–81) | IPI (not described) | DRESS syndrome: (1) 4.3%) Photosensitivity reaction: (1) 4.3% Skin toxicity: (1) 4.3% Pyoderma gangraenosum-like ulceration: (1) 4.3% Acneiform rash: (3) 13% Lichenoid exanthema: (1) 4.3% Pruritus: (8) 34.8% | Hypopigmentation: (8) 34.8% Maculopapular exanthema: (3) 13% Pruritic eczema: (1) 4.3% |
| WEBER et al, 2008 [27] USA | NRCT, Phase 1/2 | 88 Group A-MD (34) Group A-SD (30) Group B (24) | Group A-MD: 59 (34–79) Group A-SD: 57(29–87) Group B: 59.5(33–80) | IPI was administered intravenously over 90 min. IPI up to 20 mg/kg (group A, SD), multiple doses up to 5 mg/kg (group A, MD), and multiple doses up to 10 mg/kg (group B) | All patients Any grade 3 or 4 Rash: (2) 2.3% Vitiligo: (1) 1.1% Group A-MD Any grade 3 or 4 Rash: 0 Vitiligo: (1) 2.9% | Group A-SD Any grade 3 or 4 Rash: 0 Vitiligo: 0 Group B Any grade 3 or 4 Rash: (2) 8.3% Vitiligo: 0 |
| WEBER et al, 2013 [37] USA | RCT, phase 1 | 59 IPI group (20) IPI-DTIC group (19) IPI–carboplatin-paclitaxel group (20) | 56 (64–36) | IPI (10mg/kg) every 3 weeks for up to 4 doses. D group DTIC (850 mg/m2) every 3 weeks. CP group Paclitaxel (175 mg/m2) and carboplatin, every 3 weeks | IPI group Any grades Rash: (16) 80% Pruritus: (11) 55% IPI- DTIC group Any grades Rash: (9) 47.4% Pruritus: (13) 68.4% | IPI–carboplatin-paclitaxel group Any grades Rash: (15) 75% Pruritus: (13) 65% Overall Any grades Rash: (43) 72.9% Pruritus: (39) 66.1% |
| WEBER et al, 2017 [56] USA | RCT, phase 3 | 906 3 mg/kg (453) 10 mg/kg (453) | Nivolumab 56 (19–83) IPI 54 (18–86) | Nivolumab at a dose of 3 mg/kg of body weight every 2 weeks or IPI at a dose of 10 mg/kg every 3 weeks for four doses and then every 12 weeks. | Nivolumab N = 453 Any grade Pruritus: (105) 23.2% Rash: (90) 19.9% Maculopapular rash (24) 5.3% Grade 3 or 4 Rash: (5) 1.1% | IPI N = 453 Any grade Pruritus: (152) 33.6% Rash: (133) 29.4% Maculopapular rash (50) 11% Grade 3 or 4 Pruritus: (5) 1.1% Rash: (14) 3.1% Maculopapular rash (9) 2% |

(*Continued*)

**Table 1.** (Continued)

| Author, Year Country | Study design | Sample size (n) | Age in years (mean and range) | Drug (dose and schedule) | Dermatological toxicities (n) % | |
|---|---|---|---|---|---|---|
| WEN et al, 2017 [57] China | Retrospective OS | 52 | 53 (20–78) | IPI (n = 14) (intravenously at a dose of 3 mg/kg every 3 weeks, for 4 cycles) Pembrolizumab (n = 28) (intravenously at a dose of 2 mg/kg every 3 weeks, for 4 cycles) Pembrolizumab plus IPI (n = 10) (IPI intravenously at a dose of 3 mg/kg + Pembrolizumab 1 mg/kg, every 3 weeks, for 4 cycles) | IPI (n = 14) Grade 1 or 2 Pruritus: (4) 29% Rash: (3) 21% Pembrolizumab (n = 28) Grade 1 or 2 Pruritus: (3) 11% Rash: (3) 11% Vitiligo: (5) 18% | Grade 3 or 4 Rash (1) 4% Pembrolizumab + IPI (n = 10): Grade 1 or 2 Pruritus: (5) 50% Rash: (4) 40% Vitiligo: (2) 20% |
| WOLCHOCK et al, 2013 [38] USA | NRCT | 86 Concurrent Treatment (53) Sequenced treatment (33) | Concurrent Treatment 58 (22–79) Sequenced Treatment 64 (23–89) | Concurrent Treatment Cohort 1 (0.3 mg of nivolumab and 3 mg of IPI) Cohort 2 (1 mg of nivolumab and 3 mg of IPI) Cohort 2[a] (3 mg of nivolumab and 1 mg of IPI) Cohort 3 (3 mg of nivolumab and 3 mg of IPI) Cohort 4 (10 mg of nivolumab and 3 mg of IPI) Cohort 5 (10 mg of nivolumab and 10 mg of IPI) Sequenced-regimen: Cohorts 6 and 7 (1 mg and 3 mg of nivolumab), every 2 weeks for up to 48 doses. | All patients in concurrent-Regimen All grades: Rash: (29) 55% Pruritus: (25) 47% Urticaria: (1) 2% Blister: (1) 2% Grade 3 or 4 Rash: (2) 4% Pruritus: 0 Urticaria: 0 Blister: 0 | All patients in sequenced treatment All grades: Rash: (3) 9% Pruritus: (6) 18% Vitiligo: 0 Night sweats: 0 Grade 3 or 4 Rash: 0 Pruritus: 0 Vitiligo: 0 Night sweats: 0 |
| YAMAZAKI et al, 2015 [44] Japan | NRCT, phase 2 | 20 | 62.5 (29–76) | IPI (administered intravenously every 3 weeks at a dose of 3 mg/kg) | Grade 1 or 2 Rash (7) 35 Pruritus (2) 10% Alopecia (1) 5% | |
| YAMAZAKI[a] et al, 2017 [58 Japan | NRCT, phase 1b | 42 | 65 (39–89) | Pembrolizumab (administered intravenously at a dose of 2 mg/kg, every 3 weeks, during a 30-min period) | Grade 1 or 2 Pruritus: (6) 14.3% Maculopapular rash: (6) 14.3% Vitiligo (3) 7.1% Skin hypopigmentation: (2) 4.8% Dry skin (2) 4.8% | |
| YAMAZAKI[b] et al, 2017 [59] Japan | NRCT, phase 2 | 35 | 64 (28–79) | Nivolumab 2 mg/kg was given as an intravenous infusion every 3 weeks in each 6-week treatment cycle. | Any grade Leukoderma: (6) 17.1% Pruritus: (11) 31.4% Rash: (2) 5.7% Rash maculopapular: (2) 5.7% Seborrheic dermatitis: (2) 5.7% Skin hypopigmentation: (4) 11.4% | |
| YAMAZAKI[c] et al, 2017 [60] Multicentric | NRCT, phase 2 | 24 | 63 (26–81) | Nivolumab (at a dose of 3 mg/kg every 2 weeks) | Grade 1 or 2 Vitiligo: (9) 37.5% Pruritus (6) 25% Rash maculopapular: (3) 12.5% | |

(*Continued*)

**Table 1.** (Continued)

| Author, Year Country | Study design | Sample size (n) | Age in years (mean and range) | Drug (dose and schedule) | Dermatological toxicities (n) % | |
|---|---|---|---|---|---|---|
| ZIMMER[a] et al, 2015 [45] Multicentric | NRCT, phase 2 | 103 Cutaneous melanoma (83) Mucosal melanoma (7) Melanoma of unknown Primary (13) | Cutaneous melanoma 63 (29–85) Mucosal melanoma 63 (33–37) Melanoma of unknown Primary 62 (40–77) | IPI was administered intravenously over 90 min at a dose of 3 mg/kg every 3 weeks for a total of four infusions. | Grade 1 or 2 Pruritus: (11) 11% Rash: (9) 9% Erythema multiforme: (4) 5% Hand-foot-syndrome: (1) 1% Grade 3 or 4 Pruritus: 0 Rash: 0 Erythema multiforme: 0 Hand-foot-syndrome: 0 | |
| ZIMMER[b] et al, 2015 [46] Multicentric | NRCT, phase 3 | 53 | 67 (34–84) | IPI (3 mg/kg) in 3-week intervals, for four cycles. | All grades Rash: (3) 6% Pruritus: (5) 9% Erythema multiforme: (3) 6% | Grade 3 or 4 Rash: 0 Pruritus: 0 Erytema multiforme: 0 |

AE(s) = adverse event(s); CNS = central nervous system; CTCAE = Common Terminology Criteria for Adverse Events; DTIC = dacarbazine; Group A-MD = multiple dose (up to 5 mg/kg); Group A-SD = single dose (up to 20 mg/kg); Group B = multiple dose (up to 10 mg/kg); ICC = investigator's choice chemotherapy; FDA = Food and Drug Administration; IPI = Ipilimumab; irAE(s) = immune-related adverse event(s); kg = kilogram; $m^2$ = square meters; MD = multiple dose; mg = milligram(s); min = minute(s); NRCT = non-randomized clinical trial; OS = Observational Study; RCT = randomized clinical trial; SD = single dose; USA = United States of America.

syringometaplasia [52], Stevens-Johnson Syndrome [52], sever skin reactions; eczema, lichenoid or psoriasiform skin irritation [49], toxic skin eruption [51], urticaria [38], and xerosis [52], DRESS syndrome [36,55].

**Rash.** Rash grades 1 and 2 [35] was observed with the administration of tremelimumab. When compared to chemotherapy, the tremelimumab-group had a higher frequency of rash than the dacarbazine.

When ipilimumab was administered alone, grades 1 and 2 rash were developed. When ipilimumab was administered in combination with nivolumab [38,41,48,63–65] or in the combination ipilimumab plus pembrolizumab [55,57], patients developed all grades of rash. Maculopapular rash were also observed in patients using ipilimumab [48,51] and nivolumab [48,60,62,63]. Pruritic rash was also reported in patients using ipilimumab [51]. Studies evaluating monotherapy with nivolumab [50,54,59,60] showed grade 1 and 2 rash. However, the occurrence of grade 3 and 4 rash was less frequent [40,62].

**Pruritus.** Grades 1 and 2 of pruritus was observed when tremelimumab was evaluated alone [28,35], when ipilimumab was administered alone [27,29,31,32,34,39,44–47,51–53] or in combination with nivolumab [38,41,48,63–65] or pembrolizumab [55,57]. Monotherapy with nivolumab [50,54,59,60] or pembrolizumab [33,49,58,61] also showed grades 1 and 2 pruritus. Although less frequent, grade 3 and 4 pruritus were observed especially in the combination of nivolumab with ipilimumab [41,48,64], and in the combination of nivolumab with chemotherapy [30,40].

**Vitiligo.** Vitiligo was observed only with the administration of the anti-PD-1 class. Studies evaluating monotherapy with nivolumab and pembrolizumab showed the occurence of vitiligo grade 1 and 2 [33,49,58,61]. When in comparison with chemotherapy, the group treated with

nivolumab or pembrolizumab experienced a higher frequency of vitiligo than the group undergoing chemotherapy [40,42].

**Other toxicities.** Dry skin grade 1 and 2 was observed in patients using pembrolizumab alone [58], and in patients who were treated with pembrolizumab or nivolumab in comparison with chemotherapy [42,62]. Higher frequency of dry skin was observed on the group treated with immunotherapy.

Skin hypopigmentation was observed in patients using pembrolizumab [58], ipilimumab [36], and nivolumab [59]. Some patients using ipilimumab had acneiform rash/eruption [36,52], erythema multiforme [45,46], and DRESS syndrome (Drug Rash with Eosophilia and Systemic Symptoms) [36,55].

## Risk of bias among the studies

Among the clinical trials, 18 had low risk of bias [30,31,35,37,38,40,42,43,45–48,51,55,56,61,62,64], whereas 10 presented moderate risk of bias [27–29,33,41,44,58–60,63]. Domains related to the sample size and statistical analysis contributed to the classification of clinical trials as presenting moderate risk of bias. It was not possible to evaluate the representativeness of the sample size in multicenter studies [28,33,59,60,63]. Even among the studies conducted at a single center, the samples were small [29,41,44,58]. Moreover, it was not possible to verify whether the response rate was properly managed [28,29,33,44,60], or whether the samples were adequately analyzed [29,58]. In other studies, the statistical analysis of the data was not clearly described and was thus impossible to evaluate [27,41], the sample analysis was insufficient [44,60], and it was impossible to determine the suitability of the method for assessment of the condition [41] or the reliability of the condition measurement [27].

Among the observational studies, the risk of bias was considered low, moderate, and high in one [54], two [32,53], and eight studies [34,36,39,49,50,52,57,65], respectively. The studies were considered to have moderate risk of bias, when it was not possible to evaluate whether the sample represented the population or harm [32,53], or whether the statistical analysis of the data was performed adequately [53]. Among the studies with high risk of bias, most inconsistencies referred to the uncertain description of the response rate or lack of its management [34,36,39,49,50,52,57,65], unrepresentative and unstratified samples [34,39,49,50,52,57], lack of description of the statistical analyses [36], use of an unusual statistical analysis [39], or unclear adequacy of the statistical analysis [34].

The detailed evaluation of each study is presented in S3 File.

## Synthesis of the results

Among the included 39 studies, 34 were grouped to perform a meta-analysis. Four studies were excluded, because they did not grade the AEs (n = 3) [36,37,64,65] or because the reported ipilimumab AEs were caused by combined administration with chemotherapy (n = 1) [30].

We analyzed the dermatological toxicities which have been reported by two or more studies: rash, pruritus, vitiligo, dry skin, and erythema multiforme. Despite two or more studies had reported skin hypopigmentation and DRESS syndrome, it was not possible to perform the meta-analysis for these toxicities because they were not graduated [36,59] or were reported in different grades [36,55]. The heterogeneity among the studies that evaluated grades 1 to 4 rash, grades 1 and 2 pruritus, and grades 1 and 2 vitiligo was high (rash grades 1 and 2—$I^2$ = 96.93%, p = 0.001, CI = 0.20–0.31; rash grades 3 and 4—$I^2$ = 79.68%, p = 0.001, CI = 0.01–0.02; pruritus grades 1 and 2—$I^2$ = 87.21%, p = 0.001, CI = 0.21–0.27; and vitiligo grades 1 and 2—$I^2$ = 88.37%, p = 0.001, CI = 0.07–0.13). Therefore, we opted to use random effect models for the

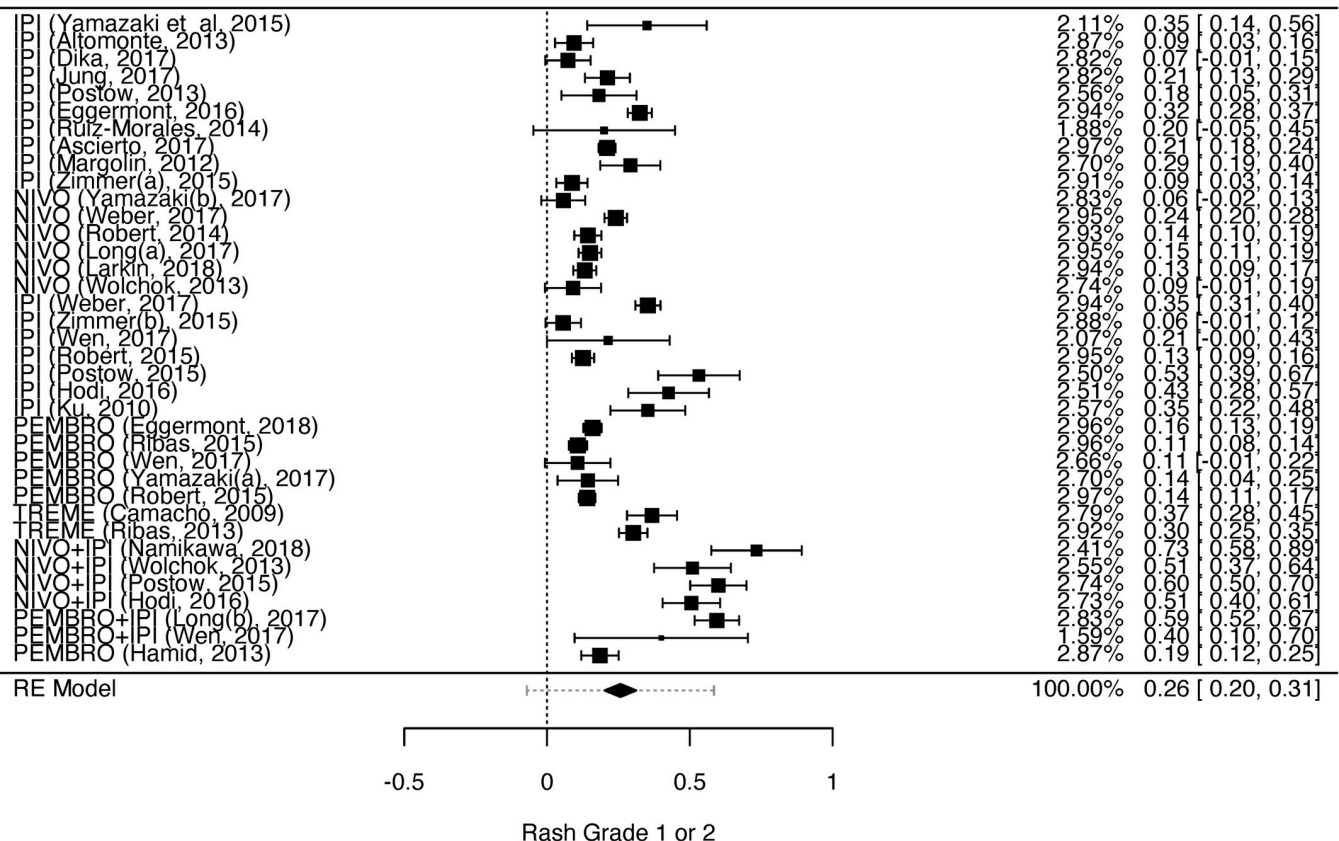

**Fig 2. Meta-analysis of rash outcome grade 1 or 2.** Elaborated by the authors using JAMOVI®. Legend: IPI—Ipilimumab, NIVO—Nivolumab, PEMBRO—Pembrolizumab, TREME—Tremelimumab.

statistical analysis. The heterogeneity could be explained by the high number of included studies (n = 39), observed by the between-study differences regarding proportion of events.

The results of the meta-analyses for grades 1 and 2 rash, pruritus, vitiligo, erythema multiforme, and dry skin are shown in Figs 2–6. The results of the meta-analysis for grades 3 and 4 rash outcomes and pruritus are shown in S1 and S2 Figs, respectively.

The prevalence of grades 1 and 2 rash and grades 3 and 4 rash was 26% (Fig 2) and 2% (S1 Fig), respectively. The prevalence of grades 1 and 2 pruritus (Fig 3) and grades 3 and 4 pruritus were 25% and 1%, respectively (S2 Fig). However, the prevalence of grades 1 and 2 vitiligo was 10% (Fig 4). We could not evaluate the prevalence of grades 3 and 4 vitiligo because only one study [27] described this event. The prevalence of erythema multiforme grade 1 and 2 was 4% (Fig 5), and dry skin grade 1 and 2 was 4% (Fig 6). The studies did not describe higher grades of erythema multiforme or dry skin. For this reason, we could not evaluate the prevalence of these events in grades 3 and 4.

## Discussion

Cutaneous AEs are often the first toxicities to occur with the use of ICIs. Despite being self-limited, these toxicities may lead to ICI dose interruption or treatment discontinuation [66]. Dermatological toxicities might be mediated by a shared antigen which is coexpressed by the tumor cells and dermoepidermal junction [10,12].

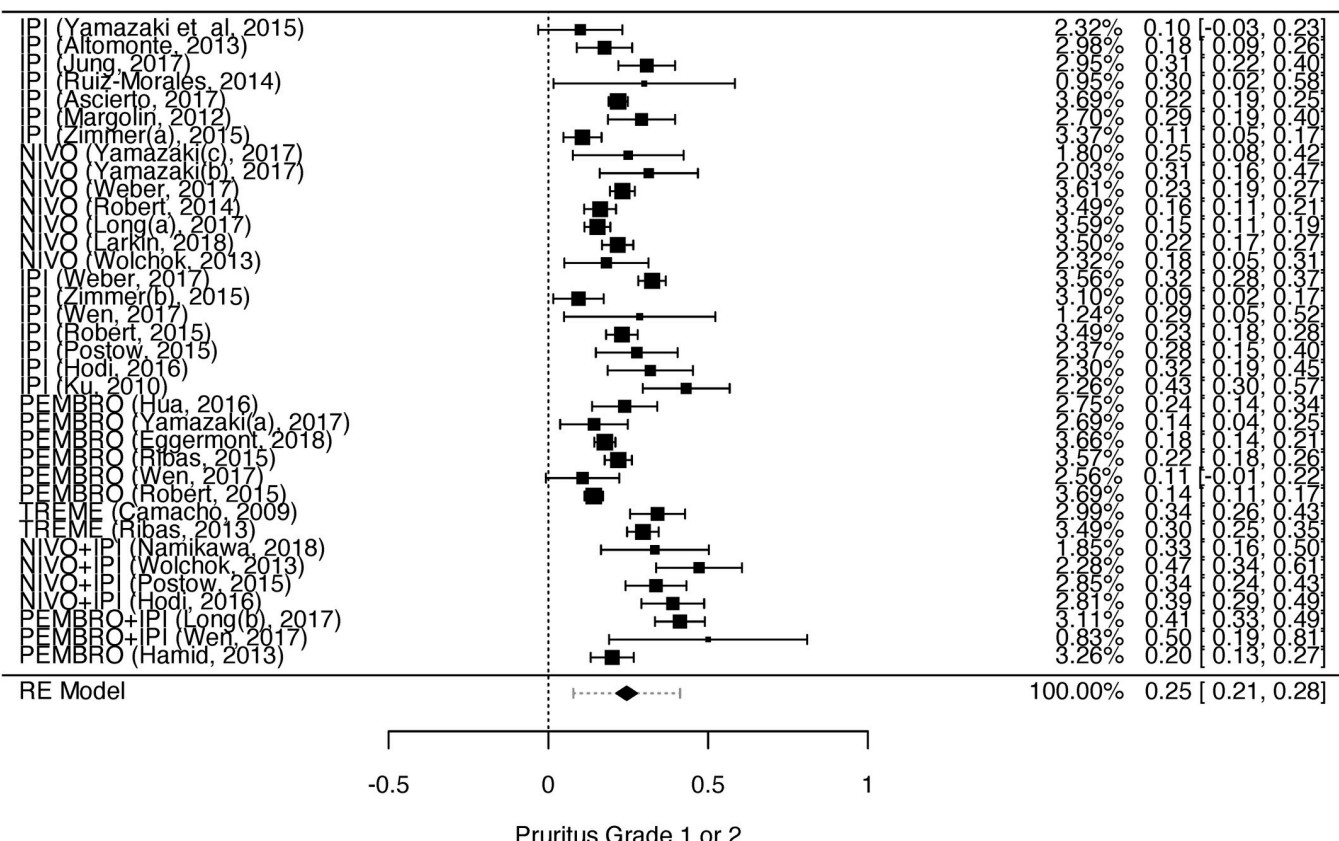

**Fig 3. Meta-analysis of pruritus outcome grade 1 or 2.** Elaborated by the authors using JAMOVI®. Legend: IPI—Ipilimumab, NIVO—Nivolumab, PEMBRO —Pembrolizumab, TREME—Tremelimumab.

Furthermore, cutaneous AEs are related to T-cell activation mediated by the blockade of PD-1 (or the PD-L1 ligand) and CTLA-4 receptors as well as the cross-reactivity between T cells directed against a tumor and T cells directed against normal tissue antigens [67]. Several hypotheses investigate factors that influence the risk of irAE development, including genetic factors, cytokines, and the composition of the patient's gastrointestinal microbiological flora [11].

The most frequent cutaneous AEs identified in this meta-analysis were grade 1 and 2 pruritus (24%), grade 1 and 2 rash (21%), grade 1 and 2 vitiligo (10%), grade 1 and 2 erythema multiforme (4%), and grade 1 and 2 dry skin. There was a higher prevalence of grades 1 and 2 compared to grades 3 and 4 of these AEs. According to Postow [68], approximately 50% of the patients treated with ICIs, mainly ipilimumab, may develop pruritus and rash. Pruritus is considered the most frequent AE reported by patients treated with ICIs. The prevalence of pruritus is high with ipilimumab administered alone or in combination with other ICIs and, usually appears concomitantly with rash; however, pruritus may also precede the rash or even appear with intact skin [67,69]. Like pruritus, rash is also one of the most frequently observed AEs in patients treated with ICIs [69] and, often start after few treatment cyles occuring mainly on the trunk and extremities [70].

Vitiligo is a commonly observed irAE with ICI treatment in patients with melanoma and it is more frequently associated with the use of PD-1 inhibitors (nivolumab and pembrolizumab) compared to the use of CTLA-4 inhibitors (ipilimumab) [71]. In this review, among the studies

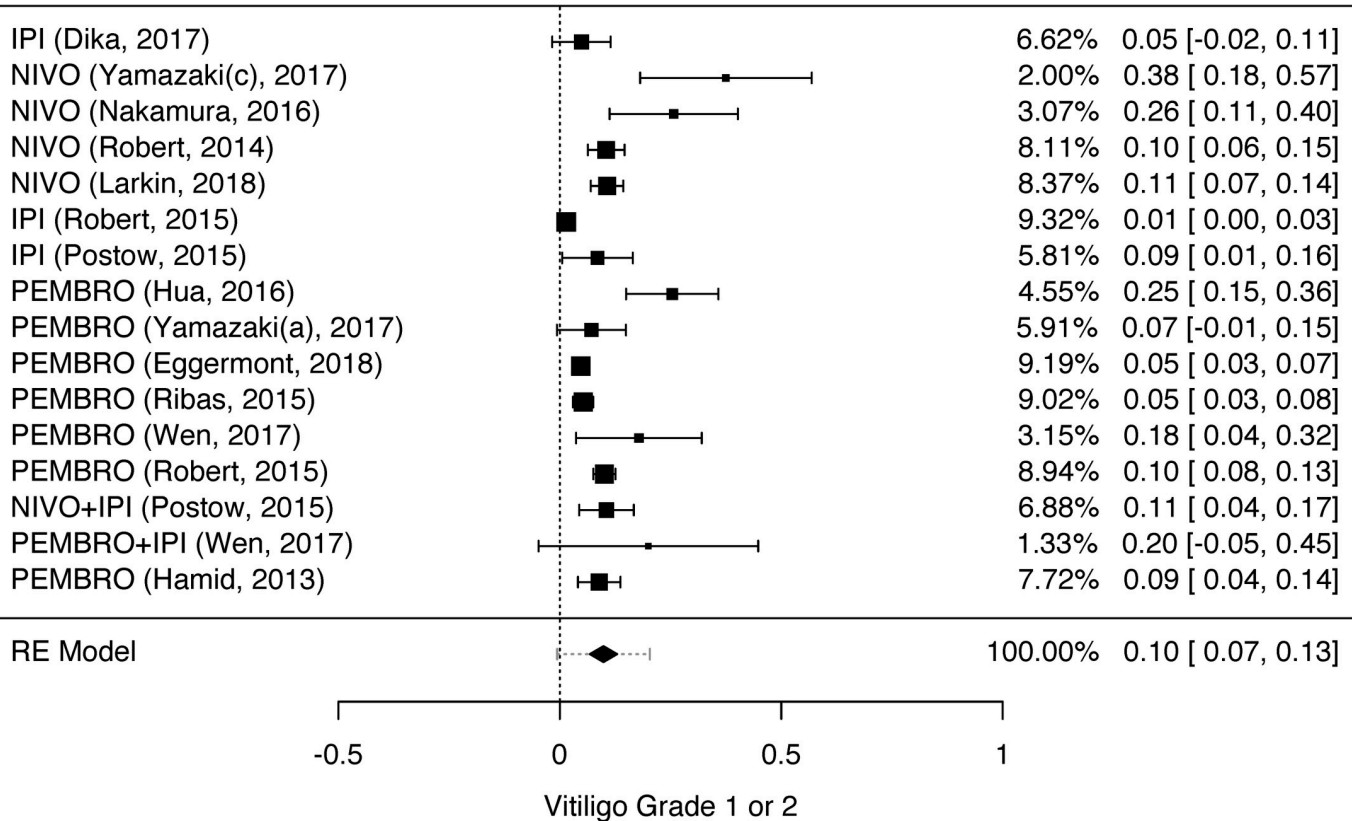

**Fig 4. Meta-analysis of vitiligo outcome grade 1 or 2.** Elaborated by the authors using JAMOVI®. Legend: IPI—Ipilimumab, NIVO—Nivolumab, PEMBRO—Pembrolizumab.

that identified vitiligo as an irAE [27,33,40–43,49,50,52,57,58,60–62], the majority of them evaluated patients treated with anti-PD-1 as a monotherapy or in combination therapy [33,40–43,49,50,57,58,60–62]. The development of vitiligo in patients treated with anti-PD-1 is caused by the activated anti-melanoma immunity that targets both malignant and healthy melanocytes [72,73] and, its occurrence has been associated with an objective response and a prolonged overall survival [12]. Studies have shown that not only vitiligo, but also rash has been associated with clinical benefits in patients treated with nivolumab [74].

Erythema multiforme (EM) is not a very commom AE observed in patients undergoing immunotherapy. However, there are some case reports showing the development of EM-related to the administration of nivolumab and ipilimumab [75,76] that required immunotherapy discontinuation, and corticosteroid treatment. Nevertheless, the studies in this review reported grade 1 and 2 EM which did not lead to treatment discontinuation. Meantime, dry skin is a commom AE related to both chemotherapy and immunotherapy. It is important to educate patients on the use of moisturizers to prevent itching.

The manifestation of cutaneous irAEs occurs at the beginning of treatment, typically 3–6 weeks after treatment initiation [77]. Most grade 3 or 4 irAEs occur later, 12–14 weeks after the beginning of treatment [78]. In this meta-analysis, the prevalence of grade 3 or 4 irAEs was lower compared to grade 1 or 2 irAEs [27,29–31,33,35,38,40,41,43,47,48,51–57,61–64]; however, more than half of the studies identified grade 3 or 4 AEs.

Most studies that presented grade 3 or 4 irAEs (n = 15) evaluated the use of ipilimumab alone [27,29,31,43,47,48,51–53] or in combination therapy [30,38,41,56,63,64]. Ipilimumab

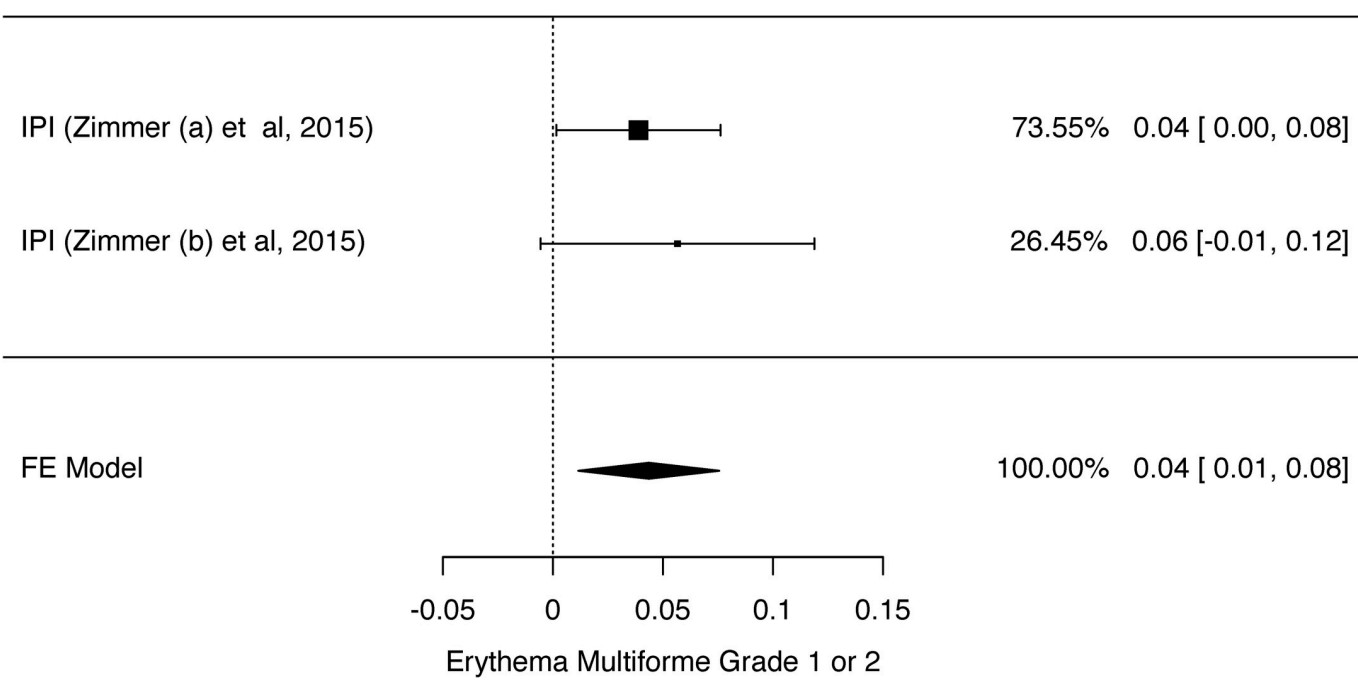

**Fig 5. Meta-analysis of erythema multiforme outcome grade 1 or 2.** Elaborated by the authors using JAMOVI®. Legend: IPI—Ipilimumab.

has a more unfavorable toxicity profile than PD-1 inhibitors. Regarding grade 3 or 4 AEs, their incidence is 20%–30% in patients receiving ipilimumab and 10%–15% in patients receiving PD-1 inhibitors. The combination of ipilimumab and PD-1 inhibitors increases the incidence of grades 3 and 4 AEs to 55% [79].

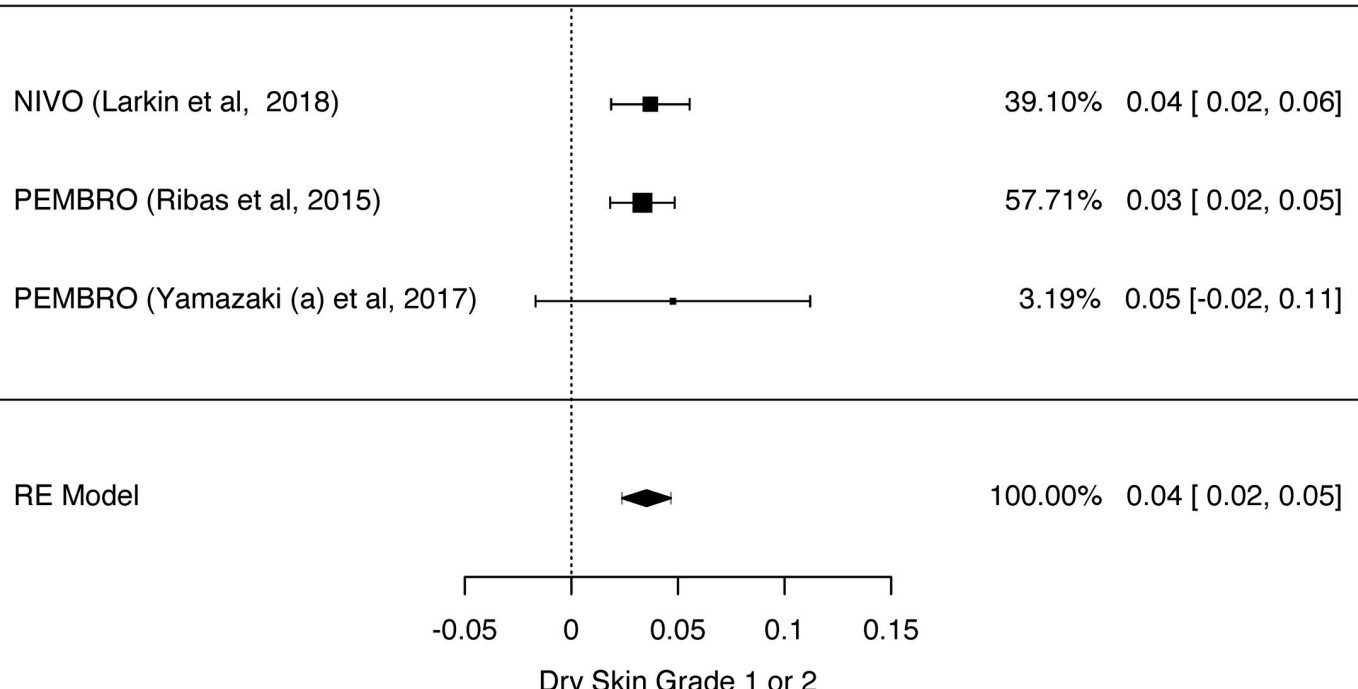

**Fig 6. Meta-analysis of dry skin outcome grade 1 or 2.** Elaborated by the authors using JAMOVI®. Legend: NIVO—Nivolumab, PEMBRO—Pembrolizumab.

Combined anti-CTLA-4 and anti-PD-1 therapy is associated with the development of more frequent cutaneous irAEs of greater severity with an earlier onset than monotherapy with ICIs [60]. Combination treatment with nivolumab and ipilimumab reportedly exacerbates their AEs and triggers high rates of grades 3 and 4 AEs. As a result, patients may repeatedly seek emergency care and may need hospitalizations and systemic immunosuppression. Further, the combination treatment has a toxicological profile that combine the side effects of both agents [63–65]. Although this combination is effective for cancer treatment, the high irEA rates are concerning [67].

Therefore, the management of adverse events requires early recognition, in addition to monitoring and classifying the grade of toxicity. The early identification of these events is critical to define the most appropriate intervention, such as treatment with the use of corticosteroids, temporary or permanent interruption of the use of ICI [80].

High-degree AEs may be potentially fatal; thus, patients should be carefully evaluated for symptoms consistent with Stevens-Johnson syndrome, pemphigus or toxic epidermal necrolysis [81]. In our review, one study identified Stevens-Johnson syndrome [52], classified as grade 4, affecting only one patient. Pemphigoid lesion grade 4 was also reported by one study [55], affecting only one patient. Though was not possible to perceive wheter these toxicities led to treatment discontinuation. However, toxic epidermal necrolysis was not listed in any of the studies. Both events are considered dermatological emergencies with high morbidity power which require an immediate intervention, hospitalization, and treatment discontinuation [82].

The frequency of irAEs increases with treatment exposure and requires long-term monitoring. Patients receiving PD-1 inhibitors may present with cutaneous AEs up to one year after the initiation of treatment [83]. Thus, follow-up periods longer than 12 months would be ideal to identify the incidence of irAEs. Overall 14 studies included in our review presented follow-up periods ranging from 1 to 5 years, which is considered an adequate time for the identification of irAEs [30,35,38,39,44,47,48,51,55,56,60–63].

Further, authors did not always report the irAEs in the most appropriate manner, since important information such as the classification [36,65] and type of scales used to classify the AEs [27,36,65] was often not reported.

A positive aspect of the irAE report was the use of the Commom Toxicity Criteria for Adverse Events (CTC-AE) scale [28–35,37–64] to classify and grade irAEs by most of the studies included in this review. The CTC-AE scale was developed to report AEs manifested by patients participating in oncological clinical trials [84]. The use of the same scale by most of the studies allowed the standardization of the irAEs reported in this review. However, Spain & Larkin [79] point out that even though the CTC-AE scale is a good scale that can also be used in clinical practice, it may lead researchers and professionals to underestimate some of the irAEs, such as pituitary gland dysfunction, because it is a scale created for a pre-immunotherapy time.

Other dermatological manifestations were identified in the studies included in this review such as maculopapular exanthema, erythema multiforme, dermatitis, acneiform rash, lichenoid exanthema, folliculitis, rosacea, eczema, leukoderma, seborrheic dermatitis, and alopecia. The majority of them were classified as grade 1 and 2.

## Limitations

The heterogeneity of the samples, lack of grading of the AEs in some studies, and unknown time of onset of the AEs may have impaired the analysis of the outcomes. It was not possible to identify subsets of patients with high possibility of displaying cutaneous side effects. Thus, future studies should classify and grade the immunomediated AEs accurately, present the time

of onset of the manifested AEs, and report the management and reversibility of the described AEs.

## Conclusion

The results of this review and meta-analysis show that the most prevalent irAEs are pruritus and rash. Even though mild and moderate irAEs were reported more frequently than severe irAEs in the included studies, there was also a significant representation of more severe AEs. Grade 3 or 4 irAEs have been associated with the use of ipilimumab. Although it is possible to manage these AEs in most cases, early identification plays a key role in the prevention of severe cases, avoiding treatment interruption.

## Supporting information

**S1 Checklist. PRISMA checklist.**
(DOC)

**S1 Fig. Meta-analysis of grade 3 or 4 rash outcomes.**
(TIF)

**S2 Fig. Meta-analysis of grade 3 or 4 pruritus outcomes.**
(TIF)

**S1 File. Search Strategy performed in the databases CINAHL, COCHRANE CENTRAL, LILACS, LIVIVO, PUBMED, SCOPUS, WEB OF SCIENCE, GOOGLE SCHOLAR, and OPENGRAY.**
(DOCX)

**S2 File. Articles excluded after reading the full text (phase 2).**
(DOCX)

**S3 File. Risk of bias in the included studies.**
(DOCX)

**S1 Protocol. PROSPERO protocol.**
(PDF)

## Author Contributions

**Conceptualization:** Náthali Felícia Mineiro dos Santos Garrett, Christiane Inocêncio Vasques.

**Data curation:** Christiane Inocêncio Vasques.

**Formal analysis:** Náthali Felícia Mineiro dos Santos Garrett, Ana Cristina Carvalho da Costa, Christiane Inocêncio Vasques.

**Funding acquisition:** Christiane Inocêncio Vasques.

**Investigation:** Náthali Felícia Mineiro dos Santos Garrett, Ana Cristina Carvalho da Costa, Christiane Inocêncio Vasques.

**Methodology:** Náthali Felícia Mineiro dos Santos Garrett, Elaine Barros Ferreira, Paula Elaine Diniz dos Reis, Christiane Inocêncio Vasques.

**Supervision:** Christiane Inocêncio Vasques.

**Writing – original draft:** Náthali Felícia Mineiro dos Santos Garrett, Ana Cristina Carvalho da Costa.

**Writing – review & editing:** Náthali Felícia Mineiro dos Santos Garrett, Ana Cristina Carvalho da Costa, Elaine Barros Ferreira, Giovanni Damiani, Paula Elaine Diniz dos Reis, Christiane Inocêncio Vasques.

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
