## [Decision Letter · Decision Letter 0]

13 Oct 2020

PONE-D-19-20870

Prevalence of dermatological toxicities in cancer patients undergoing immunotherapy: systematic review and meta-analysis

PLOS ONE

Dear Dr. Vasques:

Thank you for submitting your manuscript to PLOS ONE. After careful consideration, we feel that it has merit but does not fully meet PLOS ONE’s publication criteria as it currently stands. Therefore, we invite you to submit a revised version of the manuscript that addresses the points raised during the review process.

The reviewers had concerns listed below.  In particular concerns about the literature cited, grammatical changes, and the specific wording changes should be addressed.

We look forward to receiving your revised manuscript.

Kind regards,

Gayle E. Woloschak, PhD

Academic Editor

PLOS ONE

Additional Editor Comments:

One reviewer rejected this, the other suggested major revisions. In particular, comments were made about the need to update the literature, to take the entire literature in consideration when writing the document, and the need to correct grammatical errors. Please address these in a revision if it is possible.

Journal Requirements:

3. Please translate fully the tables you included. (E.g. S3 file has words in Portuguese.)

5. We note that this manuscript is a systematic review or meta-analysis; our author guidelines therefore require that you use PRISMA guidance to help improve reporting quality of this type of study. Please upload copies of the completed PRISMA checklist as Supporting Information with a file name “PRISMA checklist”.

Reviewers' comments:

Reviewer's Responses to Questions

**Comments to the Author**

1. Is the manuscript technically sound, and do the data support the conclusions?

Reviewer #1: Partly

Reviewer #2: Partly

2. Has the statistical analysis been performed appropriately and rigorously? 

Reviewer #1: I Don't Know

Reviewer #2: Yes

3. Have the authors made all data underlying the findings in their manuscript fully available?

Reviewer #1: No

Reviewer #2: Yes

4. Is the manuscript presented in an intelligible fashion and written in standard English?

Reviewer #1: Yes

Reviewer #2: Yes

5. Review Comments to the Author

Reviewer #1: This meta analysis of studies reported 2008-18 seeks to distill the cutaneous toxicities of ICI given alone or in combination. The review is reasonably comprehensive but spans both adjuvant and advanced disease studies of single agents and combinations, as utilized for the therapy of cutaneous melanoma. There has been an increasing literature related to the mechanism of ICI, and the immunological basis of autoimmune toxicities across a range of other organs that is not addressed in this work. Indeed, the dissection of cutaneous toxicities without prospective specification of whether dermatologic evaluations by specialists, or using biopsy pathology, is not addressed in these studies. The interpretation that rash, pruritis and vitiligo are induced by these agents (and other immunotherapies of melanoma) is not new, nor are new insights provided.

The paper is riddled with grammatical and spelling errors too numerous to recount.

Reviewer #2: Dear Editor,

Thank you for trusting me as a reviewer.

My comments are as follows:

Title

1. Make clear in the Title that the population was not “cancer” patients but “melanoma” patients.

Abstract:

1. The authors mention “To identify the prevalence of cutaneous toxicity in patients with melanoma on treatment with immune isolated checkpoint inhibitors, combined or associated with chemotherapy and/or radiotherapy.”

Did you consider checkpoint inhibitors as monotherapy AND combined with chemo and radiotherapy, or did you consider only the combinations? Please clarify.

Introduction

1. S64 : has -> have

2. S72: differente -> different

Material methods

1. PRISMA: OK

2. PROSPERO: OK

3. Inclusion και exclusion criteria: clear and analytic

4. Search strategy – study selection – data extraction: OK

5. S145: omit parenthesis

6. Risk of bias: ΟΚ

7. Summary and synthesis of the results: the authors mention the primary outcome and define it (159: «the frequence … ICI”), but in synthesis they do not mention how did they estimate the prevalence (is it pooled prevalence?). Furthermore they do not mention if they estimated confidence intervals, prevalence and CI, especially, in the cases of very low incidence (logit or double arcsine transformation method is preferable, Barendregt J et al. Meta-analysis of prevalence). In my view, even if it was not performed, it should be mentioned in the statistical analysis section.

8. Two major issues:

• Initially, the authors mention they estimated heterogeneity (I2) and based on this the used the corresponding models (random ή fixed), but they do not analyze the reasons of heterogeneity, not even in the section they present their results (s302 – 308). In addition they do not mention if there were subgroup analyses for the investigation of heterogeneity, especially when found very high (Ι2 = 88%. Ι2= 84%). Even if not performed (or when performed did not change the result?), it would be interesting to know which was the conclusion.

• There are no comments or test of publication bias (no funnel plot). This must be included.

Results

1. Flow chart – study characteristics: OK

2. S224-5: reason for not analyzed the antiPDL1. It is fine that the authors mention that they did not study this class of drugs, but I would like them to give reasons for this decision.

3. Graphic s4. In the text the authors mention that pruritus 3-4 is 1% and eruption 3-4 is 1% with reference to s4. However, in s4 the authors have included only pruritus.

4. Graphic s4. Which was the methodology the authors used for the investigation of heterogeneity among the diverse studies for pruritus and the eruption? Was it random or fixed? Please clarify. If they did so, exclusively for pruritus, they must change s315.

5. S394 differente -> different

General comments

1. Page 13,14 (s 230 – 263) the authors give the rates of AE only as a range. The latter was not pre-defined in the statistical analysis section. I presume these are the CIs, but I consider they should be defined for clarification reasons.

2. In addition, they give just the range, without a certain rate in conjunction with the range.

3. In the end of each paragraph (e.g. 234-235 or 246-247 or 257-259, there are no rates and CIs, but only their conclusions. I recommend to include them, too.

Sincerely,

Zoe Apalla

6. PLOS authors have the option to publish the peer review history of their article (what does this mean?). If published, this will include your full peer review and any attached files.

Reviewer #1: No

Reviewer #2: **Yes: **Zoe Apalla

---

## [Author Response · Author response to Decision Letter 0]

3 Nov 2020

October 14th, 2020

Gayle E. Woloschak, PhD

Academic Editor

PLOS ONE

Dear Dr. Woloschak,

Thank you for the opportunity to revise and improve the manuscript PONE-D-19-20870 entitled " Prevalence of dermatological toxicities in cancer patients undergoing immunotherapy: systematic review and meta-analysis". 

We would like to thank the reviewers for their valuable comments and suggestions. We have addressed all points raised by the reviewers and made revisions accordingly. Below, we have listed the reviewer’s comments and our responses. The changes have been highlighted in the revised version and marked in red font.

Responses to the Reviewers

Title 

1. Make clear in the Title that the population was not “cancer” patients but “melanoma” patients.

Response: We replaced the word “cancer” for “melanoma”. The modified title is “Prevalence of dermatological toxicities in patients with melanoma undergoing immunotherapy: systematic review and meta-analysis”

Abstract: 

1. The authors mention “To identify the prevalence of cutaneous toxicity in patients with melanoma on treatment with immune isolated checkpoint inhibitors, combined or associated with chemotherapy and/or radiotherapy.” 

Did you consider checkpoint inhibitors as monotherapy AND combined with chemo and radiotherapy, or did you consider only the combinations? Please clarify.

Response: It was clarified in the abstract and at the end of introduction.

"To identify the prevalence of cutaneous toxicity in patients with melanoma on treatment with immune checkpoint inhibitors as monotherapy and/or, combined with chemotherapy and/or radiotherapy" (page 2, line 38; page 4, line 82).

Introduction

1. S64 : has -> have

2. S72: differente -> different

Response: Thank you. We replaced the words.

Material methods 

1. S145: omit parenthesis

Response: Done.

2. Summary and synthesis of the results: the authors mention the primary outcome and define it (159: «the frequence … ICI”), but in synthesis they do not mention how did they estimate the prevalence (is it pooled prevalence?). Furthermore, they do not mention if they estimated confidence intervals, prevalence and CI, especially, in the cases of very low incidence (logit or double arcsine transformation method is preferable, Barendregt J et al. Meta-analysis of prevalence). In my view, even if it was not performed, it should be mentioned in the statistical analysis section. 

Response: Thank you for this appointment. The prevalence was estimated by the number of events out of the total of the sample. We add this information in the statistical analysis section (page 7, line 164).

3. Two major issues:

• Initially, the authors mention they estimated heterogeneity (I2) and based on this the used the corresponding models (random ή fixed), but they do not analyze the reasons of heterogeneity, not even in the section they present their results (s302 – 308). In addition, they do not mention if there were subgroup analyses for the investigation of heterogeneity, especially when found very high (Ι2 = 88%. Ι2= 84%). Even if not performed (or when performed did not change the result?), it would be interesting to know which was the conclusion. 

Response: We added more information to the Method Section (page 7, lines 170-172). In the synthesis of the results, we add more details to the outcomes, such as the p value and CI for each subgroup meta-analysis (page 31, lines 317-323).

• There are no comments or test of publication bias (no funnel plot). This must be included. 

Response: We performed the funnel plot and doi plot for each outcome. However, Furuya-kanamori et al (2018) recommend that funnel plot should not be considered, since in prevalence studies the graphic results could be noninterpretability. Based on this paper, we did not presented the funnel plot and doi plot. We added this information in the method section (page 7, lines 170 -172).

Results 

1. S224-5: reason for not analyzed the antiPDL1. It is fine that the authors mention that they did not study this class of drugs, but I would like them to give reasons for this decision.

Response: Some studies which evaluated anti-PDL1 drugs were identified in the database search. However, they were excluded on phase 2 for the following reasons (as mentioned in the Appendix 2 - S2 file): absence of dermatological toxicity, complementary studies with duplicated data, data extraction not possible and, other types of cancer.

2. Graphic s4. In the text the authors mention that pruritus 3-4 is 1% and eruption 3-4 is 1% with reference to s4. However, in s4 the authors have included only pruritus. 

Response: We replaced the S4 figure which now presents the results for the rash grade 3-4.

3. Graphic s4. Which was the methodology the authors used for the investigation of heterogeneity among the diverse studies for pruritus and the eruption? Was it random or fixed? Please clarify. If they did so, exclusively for pruritus, they must change s315.

Response: Thank you for this important comment about the meta-analysis effect. We used only random effect and we changed the figure.

4. S394 differente -> different

Response: Done.

General comments

1. Page 13,14 (s 230 – 263) the authors give the rates of AE only as a range. The latter was not pre-defined in the statistical analysis section. I presume these are the CIs, but I consider they should be defined for clarification reasons. 

Response: The range are not based on the CIs, it is the frequency of the event, i.e. the percentage of the events out of the total n.

2. In addition, they give just the range, without a certain rate in conjunction with the range. 

Response: As those symptoms were secondary outcomes in most individual studies, it was only possible to extract the percentage of the number of events of the AE.

3. In the end of each paragraph (e.g. 234-235 or 246-247 or 257-259, there are no rates and CIs, but only their conclusions. I recommend to include them, too. 

Response: We included all the rates in the conclusion sentence made in each paragraph (page 13, lines 226-228; page 14, lines 245-248; page 14, lines 259-271; page 15, lines 276-277).

---

## [Decision Letter · Decision Letter 1]

13 Jan 2021

PONE-D-19-20870R1

Prevalence of dermatological toxicities in patients with melanoma undergoing immunotherapy: systematic review and meta-analysis

PLOS ONE

Dear Dr. Vasques:

Thank you for submitting your manuscript to PLOS ONE. After careful consideration, we feel that it has merit but does not fully meet PLOS ONE’s publication criteria as it currently stands. Therefore, we invite you to submit a revised version of the manuscript that addresses the points raised during the review process.

One reviewer suggested revisions, the other had no changes. Please address changes in a revision.

We look forward to receiving your revised manuscript.

Kind regards,

Gayle E. Woloschak, PhD

Academic Editor

PLOS ONE

Additional Editor Comments (if provided):

One reviewer had major revisions, the other had no changes. Please address concerns in a revision.

Reviewers' comments:

Reviewer's Responses to Questions

**Comments to the Author**

1. If the authors have adequately addressed your comments raised in a previous round of review and you feel that this manuscript is now acceptable for publication, you may indicate that here to bypass the “Comments to the Author” section, enter your conflict of interest statement in the “Confidential to Editor” section, and submit your "Accept" recommendation.

Reviewer #2: All comments have been addressed

Reviewer #3: (No Response)

2. Is the manuscript technically sound, and do the data support the conclusions?

Reviewer #2: Yes

Reviewer #3: Partly

3. Has the statistical analysis been performed appropriately and rigorously? 

Reviewer #2: Yes

Reviewer #3: I Don't Know

4. Have the authors made all data underlying the findings in their manuscript fully available?

Reviewer #2: Yes

Reviewer #3: Yes

5. Is the manuscript presented in an intelligible fashion and written in standard English?

Reviewer #2: Yes

Reviewer #3: No

6. Review Comments to the Author

Reviewer #2: (No Response)

Reviewer #3: Overview: Important meta-analysis on a very common toxicity seen with ICIs. However I have some concerns about the way the information is presented. It reads a bit confusing/complicated. Another general important concept to address is WHY cutaneous toxicity in melanoma patients getting ICI is of specific interest. Notably vitiligo. Arguably the rash and vitiligo are a type of on target tox since melanoma starts as cancer of skin and attach of melanocytes suggests on target mechanism. Further vitiligo has correlated with response/survival in melanoma. In addition, similar to endocrinopathies, cutaneous tox is one of the irAEs that you can often treat through and not hold treatment but just manage symptomatically - unless progresses to SJS or TEN. Though not included in the meta-analysis I would add a paragraph just listing what has been reported in the literature when it comes to types of rashes (maculo-papular, psoriatic like, lichen planus, pemphigous etc).

Abstract:

Page 2 Line 45 - The first thing that jumped out to be was that the overall prevalence of pruritis (17%) and rash (12%) seem VERY low. I think there may be issue here with how this was arrived at. If you go to the the table 2, any grade or grade 1-2 are often higher than this. For example in your TABLE when listing reference 53 which is LONG et al JAMA Oncology 2017 if you go to the original text there is a "skin tox" listed as irAE which needs to be included and then your 11% would be more like 50%+. Further unsure why here just selected the non-TBP group.

Introduction:

Page 3 Line 63. Would reword sentence to read "The side effects related to ICIs are labeled as immune related adverse events (irAEs) and are thought to be related to [insert mechanism hypothesis]. Then would add something like "Though one can see irAes involving all body systems, cutaneous toxicity is of particular interest".

Page 3 Like 69: Would expand this to address what the cost is related to.

Eligibility

Page 4 Line 93: Would remind readers that this is MELANOMA patients.

Study Selection

Page 6 Line 125: Would replace argumentation with discussion

Comment on Statistics Methods/Synthesis/Results: I am not a statistician but this reads very confusing to me. Should somehow be simplified.

Table 1 and 2: This needs to be simplified. There is redundancy in how reported and too many words. You could consolidate to one table with columns listing Study Author and Year / Study Design [also include line of therapy here] / Drugs tested / Dermatologic Toxicities. Purpose of this paper is to highlight derm tox, other info not needed. Also I think the percentages of derm toxicities for some of the studies is not correct. See above concern about study [53].

Discussion

Page 32 Lines 322-329. This is contradictory. Say mechanism unknown but then list a mechanism. Would rephrase.

Overall would reword discussion. It is currently a repeat of what is already noted. Would comment on what is mentioned above in overview section.

7. PLOS authors have the option to publish the peer review history of their article (what does this mean?). If published, this will include your full peer review and any attached files.

Reviewer #2: **Yes: **Zoe Apalla

Reviewer #3: No

---

## [Author Response · Author response to Decision Letter 1]

18 Mar 2021

Gayle E. Woloschak, PhD

Academic Editor

PLOS ONE

Dear Dr. Woloschak,

Thank you for the opportunity to revise and improve the manuscript PONE-D-19-20870R1 entitled " Prevalence of dermatological toxicities in cancer patients undergoing immunotherapy: systematic review and meta-analysis". 

We would like to thank the reviewers for their valuable comments and suggestions. We have addressed all points raised by the reviewers and made revisions accordingly. Below, we have listed the reviewer’s comments and our responses. The changes have been highlighted in the revised version and marked in red font.

Responses to the Reviewers

Reviewer #3: 

Overview: Important meta-analysis on a very common toxicity seen with ICIs. However I have some concerns about the way the information is presented. It reads a bit confusing/complicated. Another general important concept to address is WHY cutaneous toxicity in melanoma patients getting ICI is of specific interest. Notably vitiligo. Arguably the rash and vitiligo are a type of on target tox since melanoma starts as cancer of skin and attach of melanocytes suggests on target mechanism. Further vitiligo has correlated with response/survival in melanoma. In addition, similar to endocrinopathies, cutaneous tox is one of the irAEs that you can often treat through and not hold treatment but just manage symptomatically - unless progresses to SJS or TEN. Though not included in the meta-analysis I would add a paragraph just listing what has been reported in the literature when it comes to types of rashes (maculo-papular, psoriatic like, lichen planus, pemphigous etc).

Response: As suggested by the reviewer, we added on discussion section a paragraph listing other dermatological toxicities mentioned by the studies included in the review. “Other dermatological manifestations were identified in the studies included in this review such as maculopapular exanthema, erythema multiforme, dermatitis, acneiform rash, lichenoid exanthema, folliculitis, rosacea, eczema, leukoderma, seborrheic dermatitis, and alopecia. The majority of them were classified as grade 1 and 2. Although it is possible to manage these AEs in most cases, early identification plays a key role in the prevention of severe cases, avoiding treatment interruption.” (page 32, lines 408-413)

Abstract:

Page 2 Line 45 - The first thing that jumped out to be was that the overall prevalence of pruritis (17%) and rash (12%) seem VERY low. I think there may be issue here with how this was arrived at. If you go to the the table 2, any grade or grade 1-2 are often higher than this. For example in your TABLE when listing reference 53 which is LONG et al JAMA Oncology 2017 if you go to the original text there is a "skin tox" listed as irAE which needs to be included and then your 11% would be more like 50%+. Further unsure why here just selected the non-TBP group.

Response: We decided not to present the overall prevalence for pruritus and rash. These signs/symptoms are adverse events that evolve gradually, for example, the patient that presents pruritus classified as grade 1 and 2 in the beginning of the treatment can evolve to grade 3 and 4 later. Thus, when calculating the overall prevalence we may be counting the same patient twice. For this reason, we decided to present only the results of the subgroup analysis grade 1 and 2 and, grade 3 and 4 for both pruritus and rash.

The modified sentence on the abstract is : “The results suggest that the most prevalent side effect was grade 1 and 2 pruritus (24%), followed by grade 1 and 2 rash (21%) and grade 1 and 2 vitiligo (10%).” (page 2, lines 44-46)

Regarding the reference n. 53 (Long et al, 2017), which now is n. 54 due to the insertion of a new reference in the introduction, we inform that we did not selected one specific group. The data from both non-TBP group and TBP group have already been described in table. As suggested by the reviewer, we included the data related to “skin tox” in table 1. 

Introduction:

Page 3 Line 63. Would reword sentence to read "The side effects related to ICIs are labeled as immune related adverse events (irAEs) and are thought to be related to [insert mechanism hypothesis]. Then would add something like "Though one can see irAes involving all body systems, cutaneous toxicity is of particular interest".

Response: We inserted the reviewer’s suggestion. The modified sentence is presented above.

“(…) The side effects related to ICIs are labeled as immune-related adverse events (irAEs) and are thought to be related to the inflammatory response caused in several organs due to the stimulation of the immune system, especially of T cells [9,10, 11]. Though one can see irAEs involving all body systems, cutaneous toxicity is of particular interest.” (page 3, lines 63-67)

Page 3 Like 69: Would expand this to address what the cost is related to.

Response: We included the information as suggested.

“The cost associated with the management of dermatological toxicity in patients with metastatic melanoma reaches US $ 21,726.00 per month, which represent the total adjudicated amount paid to all providers for inpatient and outpatient services and drugs [14].” (page 4, lines 77-79)

Eligibility

Page 4 Line 93: Would remind readers that this is MELANOMA patients.

Response: We included the word melanoma. 

“We included clinical trials (randomized and non-randomized) and observational studies that evaluated melanoma cancer patients undergoing treatment with a single ICI, a combination of ICIs, or a combination of an ICI with chemotherapy and/or radiotherapy and that described the prevalence of dermatological toxicity.” (page 5, lines 105)

Study Selection

Page 6 Line 125: Would replace argumentation with discussion

Response: We replaced the word argumentation for discussion. 

“(…) Any disagreements in the first or second phase were resolved by discussion until an agreement was reached between the two authors.(…)” (page 6, line 136)

Table 1 and 2: This needs to be simplified. There is redundancy in how reported and too many words. You could consolidate to one table with columns listing Study Author and Year / Study Design [also include line of therapy here] / Drugs tested / Dermatologic Toxicities. Purpose of this paper is to highlight derm tox, other info not needed. Also I think the percentages of derm toxicities for some of the studies is not correct. See above concern about study [53].

Response: We consolidated the tables. Table 2 now is presented as table 1 in which we inserted the columns author/year/contry and, study design that were originally from table 1. All other data have not been changed. Regarding the study mentioned (reference n. 53, that now is 54 because of the insertion of one reference in the introduction), the percentages are correct. We inserted only the acronymun TBP (treatment beyond progression) and non-TBP to be the same as described by the author in the article, but all the data about non-TBP and TBP groups had already been described since the first version of the manuscript. (page 13)

STUDY CHARACTERISTICS SAMPLE CHARACTERISTICS EXPOSURE CHARACTERISTICS OUTCOME CHARACTERISTICS

Author, Year

Country Study design Sample size (n) Age in years (mean and range) Drug (dose and schedule) Duration of treatment Follow-up Dermatological toxicities (n) % Main Conclusions

LONGa et al, 2017 [54]

Australia Retrospective OS 306 

Non-beyond

progression

(221)

Beyond

progression

(85)

 62 (18-90) Nivolumab (3mg/kg every 2 weeks) Until progression or unacceptable toxic effects. Not described 

 Non-treatment beyond

Progression (Non-TBP)

Any Grade

Skin adverse event: (58) 26%

Pruritus: (25) 11%

Rash: (23) 10%

Grade 3 or 4

Skin adverse event: (2) 1%

Pruritus: (1) <1%

Rash: 0 Treatment Beyond

Progression (TBP)

Any Grade

Skin adverse event: (43) 51%

Pruritus: (23) 27%

Rash: (23) 27%

Grade 3 or 4 

Skin adverse event: (1) 1%

Pruritus: 0

Rash: 0 Patients treated beyond

their first disease progression can experience a tumor response with continued nivolumab treatment, with a safety profile consistent with that observed in patients who did not receive further treatment. 

Discussion

Page 32 Lines 322-329. This is contradictory. Say mechanism unknown but then list a mechanism. Would rephrase.

Response: We rewrote the sentence. The modified sentence is “Cutaneous AEs are the first toxicities to occur with the use of ICIs. Despite being self-limited, these toxicities may lead to ICI dose reduction and treatment discontinuation [66]. Dermatological toxicities might be mediated by a shared antigen which is coexpressed by the tumor cells and dermoepidermal junction [10, 12].”

Overall would reword discussion. It is currently a repeat of what is already noted. Would comment on what is mentioned above in overview section.

Response: We reworded many aspects of the discussion in order to address what was mentioned in the overview section. All the changes we made are highlighted in red font.

---

## [Decision Letter · Decision Letter 2]

16 Apr 2021

PONE-D-19-20870R2

Prevalence of dermatological toxicities in patients with melanoma undergoing immunotherapy: systematic review and meta-analysis

PLOS ONE

Dear Dr. Vasques:

Thank you for submitting your manuscript to PLOS ONE. After careful consideration, we feel that it has merit but does not fully meet PLOS ONE’s publication criteria as it currently stands. Therefore, we invite you to submit a revised version of the manuscript that addresses the points raised during the review process.

The reviewers considered that changes made were not yet sufficient for the work.

We look forward to receiving your revised manuscript.

Kind regards,

Gayle E. Woloschak, PhD

Academic Editor

PLOS ONE

Reviewers' comments:

Reviewer's Responses to Questions

**Comments to the Author**

1. If the authors have adequately addressed your comments raised in a previous round of review and you feel that this manuscript is now acceptable for publication, you may indicate that here to bypass the “Comments to the Author” section, enter your conflict of interest statement in the “Confidential to Editor” section, and submit your "Accept" recommendation.

Reviewer #3: (No Response)

2. Is the manuscript technically sound, and do the data support the conclusions?

Reviewer #3: Yes

3. Has the statistical analysis been performed appropriately and rigorously? 

Reviewer #3: N/A

4. Have the authors made all data underlying the findings in their manuscript fully available?

Reviewer #3: Yes

5. Is the manuscript presented in an intelligible fashion and written in standard English?

Reviewer #3: Yes

6. Review Comments to the Author

Reviewer #3: I still think you miss the opportunity to make this a more complete meta-analysis of dermatologic toxicities. As it stand right now it is mostly a summary of the incidence of "rash" "pruritis" and "vitiligo". Though as you can see a number of studies are more detailed in their descriptions. You hit on this in the conclusion a bit but the majority of text just summarizes incidence of "rash" and "pruritis" which is very common and actually in clinical practice not always clinically significant. It is the more specific toxicities such as pemphigus, EN, etc that you actually hold or stop drug.

I would maybe like to see the paper organized by type of skin toxicity as opposed to results of individual studies. A number of the studies detail other more clinical significant cutaneous toxicities. The Table is still a bit long and too detailed. Would remove the column on duration of treatment as well as main conclusion (purpose of paper is not to educate on outcomes of the studies as pertains to cancer effect).

7. PLOS authors have the option to publish the peer review history of their article (what does this mean?). If published, this will include your full peer review and any attached files.

Reviewer #3: No

---

## [Author Response · Author response to Decision Letter 2]

28 May 2021

May 28th, 2021

Gayle E. Woloschak, PhD

Academic Editor

PLOS ONE

Dear Dr. Woloschak,

Thank you for the opportunity to revise and improve the manuscript PONE-D-19-20870R1 entitled " Prevalence of dermatological toxicities in cancer patients undergoing immunotherapy: systematic review and meta-analysis". 

We would like to thank the reviewers for their valuable comments and suggestions. We have addressed all points raised by the reviewers and made revisions accordingly. Below, we have listed the reviewer’s comments and our responses. The changes have been highlighted in the revised version and marked in red font.

Responses to the Reviewers

Reviewer #3: 

I still think you miss the opportunity to make this a more complete meta-analysis of dermatologic toxicities. As it stand right now it is mostly a summary of the incidence of "rash" "pruritis" and "vitiligo". Though as you can see a number of studies are more detailed in their descriptions. You hit on this in the conclusion a bit but the majority of text just summarizes incidence of "rash" and "pruritis" which is very common and actually in clinical practice not always clinically significant. It is the more specific toxicities such as pemphigus, EN, etc that you actually hold or stop drug.

Response: As suggested by the reviewer, we looked again at the studies included in this review to bring out the dermatological toxicities in more detail, which can be seen in Table 1 (page 10-20), and in the results section (pages 22 -24, lines 247 - 317). 

We identifiy that erythema multiforme, skin hypopigmentation, dry skin and DRESS syndrome were also reported by two or more studies. So we reported this in the result section (pages 22 -24, lines 247 - 317). For these reason, we rerun the meta-analysis for rash, pruritus and vitiligo, and we run the meta-analysis for erythema multiforme and dry skin, now on Jamovi software (page 25-26, lines 349 - 358; 376-390). It was not possible to run the meta-analysis for skin hypopigmentation because the studies did not classified this toxicity. Among the studies that reported DRESS syndrome, one reported the toxicity as grade 1and 2, and the other one as grade 3 and 4, which made impossible to perform the meta-analysis. 

We also inserted a paragraph on discussion regard the inclusion of erythema multiforme and dry skin meta-analysis: “Erythema multiforme (EM) is not a very commom AE observed in patients undergoing immunotherapy. However, there are some case reports showing the development of EM-related to the administration of nivolumab and ipilimumab [75, 76] that required immunotherapy discontinuation, and corticosteroid treatment. Nevertheless, the studies in this review reported grade 1 and 2 EM which did not lead to treatment discontinuation. Meantime, dry skin is a commom AE related to both chemotherapy and immunotherapy. It is important to educate patients on the use of moisturizers to prevent itching” (page 28, lines 427-433).

Regarding the more specific toxicities with clinical impact, only one case of pemphigus (long et al, 2017), Steve-Johnson syndrome (Dika et al., 2017), and EN (Ascierto et al., 2017) were reported by the studies, but the authors did not make clear if these toxicities led to treatment discontinuation. However, we had already included a paragraph in the discussion section reporting the data about Steve-Johnson syndrome identified in this review, and stating the importance of monitoring signs and symptoms indicative of Steve-Johnson syndrome and toxic epidermal necrolisis. Now, we included a sentence about the data we found about pemphigoid lesion (page 29, lines 463-465): 

“High-degree AEs may be potentially fatal; thus, patients should be carefully evaluated for symptoms consistent with Stevens-Johnson syndrome, pemphigus or toxic epidermal necrolysis [81]. In our review, one study identified Stevens-Johnson syndrome [52], classified as grade 4, affecting only one patient. Pemphigoid lesion grade 4 was also reported by one study [55], affecting only one patient. Though was not possible to perceive wheter these toxicities led to treatment discontinuation. However, toxic epidermal necrolysis was not listed in any of the studies. Both events are considered dermatological emergencies with high morbidity power which require an immediate intervention, hospitalization, and treatment discontinuation [82].”

Reviewer #3: 

I would maybe like to see the paper organized by type of skin toxicity as opposed to results of individual studies. A number of the studies detail other more clinical significant cutaneous toxicities. The Table is still a bit long and too detailed. Would remove the column on duration of treatment as well as main conclusion (purpose of paper is not to educate on outcomes of the studies as pertains to cancer effect).

Response: We are now presenting the results by type of skin toxicity (pages 22 -24, lines 247-317).

 As suggested, we removed the column on duration of treatment and main conclusion from table 1.

---

## [Decision Letter · Decision Letter 3]

2 Jul 2021

PONE-D-19-20870R3

Prevalence of dermatological toxicities in patients with melanoma undergoing immunotherapy: systematic review and meta-analysis

PLOS ONE

Dear Dr. Vasques:

Thank you for submitting your manuscript to PLOS ONE. After careful consideration, we feel that it has merit but does not fully meet PLOS ONE’s publication criteria as it currently stands. Therefore, we invite you to submit a revised version of the manuscript that addresses the points raised during the review process.

Minor revisions have been proposed as suggested below.

We look forward to receiving your revised manuscript.

Kind regards,

Gayle E. Woloschak, PhD

Academic Editor

PLOS ONE

Journal Requirements:

Additional Editor Comments (if provided):

Some minor revisions related to rewording should be done before the next revision.

Reviewers' comments:

Reviewer's Responses to Questions

**Comments to the Author**

1. If the authors have adequately addressed your comments raised in a previous round of review and you feel that this manuscript is now acceptable for publication, you may indicate that here to bypass the “Comments to the Author” section, enter your conflict of interest statement in the “Confidential to Editor” section, and submit your "Accept" recommendation.

Reviewer #3: (No Response)

2. Is the manuscript technically sound, and do the data support the conclusions?

Reviewer #3: Yes

3. Has the statistical analysis been performed appropriately and rigorously? 

Reviewer #3: Yes

4. Have the authors made all data underlying the findings in their manuscript fully available?

Reviewer #3: Yes

5. Is the manuscript presented in an intelligible fashion and written in standard English?

Reviewer #3: Yes

6. Review Comments to the Author

Reviewer #3: 1) Page 3 Line 55: Would reword - patients achieving responses vs delay of disease progression are two different outcomes. May say something like "Many patients have been living longer due to remarkable responses and delay of disease progression"

2) Page 3 Line 63: Would reword - "Although in many cancers one can see anti-tumor activity with ICIs and traditional chemotherapy, the types, mechanisms, and rates of side effects differ"

3) Page 4 Line 87: Would reword - This review is a comprehensive report on the prevalence of dermatological toxicity in patients ...

4) Table 1: I would still further simplify. Would remove the Duration of treatment, Follow up, and main conclusions since really just highlighting incidence of cutaneous irAE

5) Page 29 Line 329: Reword - Cutaneous AEs are often the first toxicities

6) Page 29 Line 330: Reword - ... these toxicities may lead to IDI dose interruption or treatment discontinuation (there really isn't dose reduction with ICIs)

7) Page 32 Line 410: Would take sentence that starts with "Although it is possible ... and make it last sentence of conclusion

8) Page 33 Line 424: Would remove sentence that starts with "Vitiligo" and "Regarding" Make conclusion one paragraph

7. PLOS authors have the option to publish the peer review history of their article (what does this mean?). If published, this will include your full peer review and any attached files.

Reviewer #3: No

---

## [Author Response · Author response to Decision Letter 3]

14 Jul 2021

July 12th, 2021

Gayle E. Woloschak, PhD

Academic Editor

PLOS ONE

Dear Dr. Woloschak,

Thank you for the opportunity to revise and improve the manuscript PONE-D-19-20870R1 entitled " Prevalence of dermatological toxicities in cancer patients undergoing immunotherapy: systematic review and meta-analysis". 

We would like to thank the reviewers for their valuable comments and suggestions. We have addressed all points raised by the reviewers and made revisions accordingly. Below, we have listed the reviewer’s comments and our responses. The changes have been highlighted in the revised version and marked in red font.

Responses to the Reviewers

Reviewer #3: 

1) Page 3 Line 55: Would reword - patients achieving responses vs delay of disease progression are two different outcomes. May say something like "Many patients have been living longer due to remarkable responses and delay of disease progression"

Our response: We reworded the sentence as suggested by the reviewer: “Many patients have been living longer due to remarkable responses and delay of disease progression during ICI treatment [3].”

Reviewer #3: 

2) Page 3 Line 63: Would reword - "Although in many cancers one can see anti-tumor activity with ICIs and traditional chemotherapy, the types, mechanisms, and rates of side effects differ"

Our response: We reworded the sentence as suggested by the reviewer: “Although in many cancers one can see anti-tumor activity with ICIs and traditional chemotherapy, the types, mechanisms, and rates of side effects differ”. 

Reviewer #3: 

3) Page 4 Line 87: Would reword - This review is a comprehensive report on the prevalence of dermatological toxicity in patients ...

Our response: As suggested by the reviewer, we reworded the sentence: “This review is a comprehensive report on the prevalence of dermatological toxicity in patients with melanoma using ICIs as monotherapy and/or in combination with chemotherapy and/or radiotherapy.”

Reviewer #3: 

4) Table 1: I would still further simplify. Would remove the Duration of treatment, Follow up, and main conclusions since really just highlighting incidence of cutaneous irAE

Our response: We removed the follow-up column from table 1. The duration of treatment and main conclusion columns had already been removed on previous revision. The table 1 now presents the following data: author/year/country, study design, sample size, age, drug (dose and schedule) and dermatological toxicities.

Reviewer #3: 

5) Page 29 Line 329: Reword - Cutaneous AEs are often the first toxicities

Our response: We reworded as suggested: “Cutaneous AEs are often the first toxicities to occur with the use of ICIs. (…)” 

Reviewer #3: 

6) Page 29 Line 330: Reword - ... these toxicities may lead to IDI dose interruption or treatment discontinuation (there really isn't dose reduction with ICIs)

Our response: As suggested by the reviewer, we reworded the sentence: “Despite being self-limited, these toxicities may lead to ICI dose interruption or treatment discontinuation.”

Reviewer #3: 

7) Page 32 Line 410: Would take sentence that starts with "Although it is possible ... and make it last sentence of conclusion

Our response: Done.

Reviewer #3: 

8) Page 33 Line 424: Would remove sentence that starts with "Vitiligo" and "Regarding" Make conclusion one paragraph

Our response: We removed the sentences as suggested by the reviewer, included the sentence as suggested by the reviewer’s comment #7, and made the conclusion one paragraph as follow: “The results of this review and meta-analysis show that the most prevalent irAEs are pruritus and rash. Even though mild and moderate irAEs were reported more frequently than severe irAEs in the included studies, there was also a significant representation of more severe AEs. Grade 3 or 4 irAEs have been associated with the use of ipilimumab. Although it is possible to manage these AEs in most cases, early identification plays a key role in the prevention of severe cases, avoiding treatment interruption.”

---

## [Decision Letter · Decision Letter 4]

23 Jul 2021

Prevalence of dermatological toxicities in patients with melanoma undergoing immunotherapy: systematic review and meta-analysis

PONE-D-19-20870R4

Dear Dr. Vasques:

We’re pleased to inform you that your manuscript has been judged scientifically suitable for publication and will be formally accepted for publication once it meets all outstanding technical requirements.

Kind regards,

Gayle E. Woloschak, PhD

Section Editor

PLOS ONE

Additional Editor Comments (optional):

Thank you for addressing concerns raised by the review.

Reviewers' comments:

Reviewer's Responses to Questions

**Comments to the Author**

1. If the authors have adequately addressed your comments raised in a previous round of review and you feel that this manuscript is now acceptable for publication, you may indicate that here to bypass the “Comments to the Author” section, enter your conflict of interest statement in the “Confidential to Editor” section, and submit your "Accept" recommendation.

Reviewer #3: All comments have been addressed

2. Is the manuscript technically sound, and do the data support the conclusions?

Reviewer #3: Yes

3. Has the statistical analysis been performed appropriately and rigorously? 

Reviewer #3: Yes

4. Have the authors made all data underlying the findings in their manuscript fully available?

Reviewer #3: Yes

5. Is the manuscript presented in an intelligible fashion and written in standard English?

Reviewer #3: Yes

6. Review Comments to the Author

Reviewer #3: (No Response)

7. PLOS authors have the option to publish the peer review history of their article (what does this mean?). If published, this will include your full peer review and any attached files.

Reviewer #3: No

---

## [Editor Report · Acceptance letter]

29 Jul 2021

PONE-D-19-20870R4 

Prevalence of dermatological toxicities in patients with melanoma undergoing immunotherapy: systematic review and meta-analysis 

Dear Dr. Inocêncio Vasques:

I'm pleased to inform you that your manuscript has been deemed suitable for publication in PLOS ONE. Congratulations! Your manuscript is now with our production department. 

Kind regards, 

on behalf of

Dr. Gayle E. Woloschak 

Section Editor

PLOS ONE